# miR156 Is a Negative Regulator of Aluminum Response in *Medicago sativa*

**DOI:** 10.3390/plants14060958

**Published:** 2025-03-19

**Authors:** Gamalat Allam, Solihu K. Sakariyahu, Tim McDowell, Tevon A. Pitambar, Yousef Papadopoulos, Mark A. Bernards, Abdelali Hannoufa

**Affiliations:** 1Agriculture and Agri-Food Canada, 1391 Sandford Street, London, ON N5V 4T3, Canada; gamalat.allam@agr.gc.ca (G.A.); solihukayode.sakariyahu@agr.gc.ca (S.K.S.); tevon.pitambar@agr.gc.ca (T.A.P.); 2Department of Biology, University of Western Ontario, 1151 Richmond Street, London, ON N6A 3K7, Canada; bernards@uwo.ca; 3Agriculture and Agri-Food Canada, 58 River Road, Truro, NS B2N 5E3, Canada

**Keywords:** *Medicago sativa*, miR156, PG1, PG4, AUX1, PIN2, Al stress

## Abstract

Aluminum (Al) toxicity is a serious environmental constraint facing crop production in acidic soils, primarily due to the oxidative damage it causes to plant tissues. Alfalfa (*Medicago sativa*), a globally important forage crop, is highly susceptible to Al-induced stress, necessitating the development of Al-tolerant cultivars for sustainable forage production. In this study, we investigated the regulatory role of miR156 in Al stress response in alfalfa. Transcript analysis revealed significant downregulation of miR156 in alfalfa roots after 8 h of Al exposure, suggesting a negative role for miR156 in response to Al. To further investigate the role of miR156 in regulating agronomic traits and alfalfa’s Al tolerance, we utilized the short tandem target mimic (STTM) method to silence miR156 in alfalfa (MsSTTM156), which led to an upregulation of SQUAMOSA PROMOTER BINDING-LIKE (SPL) target genes, albeit with variable miR156 dose-dependent effects across different transgenic genotypes. Morphological characterization of MsSTTM156 plants revealed significant negative changes in root architecture, root and shoot biomass, as well as flowering time. Under Al stress, overexpression of miR156 in alfalfa (MsmiR156OE) resulted in stunted growth and reduced biomass, whereas moderate MsmiR156 silencing enhanced root dry weight and increased stem basal diameter. In contrast, MsmiR156OE reduced plant height, stem basal diameter, shoot branching, and overall biomass under Al stress conditions. At the molecular level, silencing miR156 modulated the transcription of cell wall-related genes linked to Al tolerance, such as polygalacturonase 1(*MsPG1)* and polygalacturonase 4 (*MsPG4)*. Furthermore, miR156 influenced the expression of indole-3-acetic acid (IAA) transport-related genes auxin transporter-like protein (*MsAUX1)* and auxin efflux carrier components 2 (*MsPIN2)*, with MsSTTM156 and MsmiR156OE plants showing lower and higher transcript levels, respectively, upon Al exposure. These findings reveal the multi-layered role of miR156 in mediating Al tolerance, providing valuable insights into the genetic strategies that regulate response to Al stress in alfalfa.

## 1. Introduction

Alfalfa (*Medicago sativa* L.) is one of the most extensively cultivated perennial forage crops for livestock feed due to its desirable traits, including rapid biomass production, high nutritional value, and an exceptionally high energy content [1,2]. Beyond these beneficial traits, its deep-rooting system not only reduces soil erosion but also enhances plant survival during drought and nutrient stress [3,4]. Alfalfa is used for crop rotations and soil enhancement because of its ability to form a symbiotic relationship with rhizobium bacteria, enhancing soil nitrogen balance through nitrogen fixation [5,6], and thus supporting sustainable agriculture practices [7]. Despite its numerous advantages, the cultivation of alfalfa is severely constrained by soil acidity and Al toxicity [8,9,10].

Al toxicity is a major abiotic stress affecting crop productivity in acidic soils, which represent more than half of the world’s arable land [11]. In acidic soils, Al exists in a soluble ionic form, Al^3+^ [12], which exhibits significant toxicity to plant growth of various species, affecting multiple cellular processes and inhibiting root elongation and nutrient uptake, leading to reduced plant growth and yield [13]. In alfalfa, Al toxicity primarily inhibits root growth, reducing biomass and limiting crop productivity [14,15]. This damage extends to physiological drought by disturbing root cell integrity, affecting water and nutrient uptake, and destabilizing membrane function, which in turn impairs ion transport crucial for physiological processes [16,17]. This damage further extends to photosynthetic inhibition by reducing chlorophyll synthesis, gas exchange, stomatal conductance, transpiration rate, electron transport, and photosystem PSI and PSII photochemical efficiency, thereby affecting plant growth and development [18,19,20]. Furthermore, Al toxicity accelerates reactive oxygen species (ROS) accumulation, leading to membrane damage, protein degradation, and programmed cell death [15,21,22]. Notably, modern agricultural practices further exacerbate this issue through the application of NH_4_^+^ or ammonium-based fertilizers, increasing soil acidity through nitrification processes, thereby worsening the risk of Al toxicity [23].

microRNA156 (miR156) is a highly conserved small RNA in plants that functions by targeting Squamosa Promoter Binding Protein-Like (SPL) transcription factors, which in turn regulate various aspects of plant growth and development [23]. These aspects include the timing of vegetative phase changes and floral induction, the rate of leaf initiation, shoot regeneration and branching, anthocyanin and trichome production, stress responses, carotenoid biosynthesis, and lateral root development [24,25,26,27,28,29]. SPLs interact with phytohormones such as gibberellins to regulate floral transition [30] and regulate root regenerative capacity partially through auxin [31]. SPL8 and SPL9 also regulate drought response in alfalfa [32,33].

miR156 has emerged as a potential biotechnological tool in crop improvement, as evidenced by its ability to modulate traits in various plant species [34]. For instance, upregulating miR156d in alfalfa resulted in delayed flowering, increased biomass production, root growth, and nodulation [35]. Expressing AtmiR156b in *Brassica napus* boosted lutein and β-carotene levels in seeds and improved the number of flowering shoots [36]. Similarly, OsmiR156b expression in switchgrass significantly increased biomass production through enhanced tiller number [37]. Conversely, miR156 overexpression in potato reduced tuber yield and altered plant architecture [38], whereas Arabidopsis plants overexpressing miR156 promoted the juvenile phase with increased shoot biomass, while miR156 silencing led to early flowering and reduced shoot biomass [39]. In tobacco, miR156a overexpression significantly altered the distribution of Cu, Zn, Mn, and Fe in plant tissues and regulated key genes involved in mineral nutrient uptake, transport, and storage, thereby maintaining nutrient homeostasis [40]. Transgenic *Nta-miR156aOE* plants accumulated more Zn in leaves and roots while maintaining unchanged levels in stems [40]. This suggests miR156 enhances Zn uptake and translocation but limits its movement to stems, making it a key regulator of Zn homeostasis [40].

Beyond its developmental roles, miR156 has emerged as a critical regulator of abiotic stress responses in plants [41]. Overexpression of miR156d in alfalfa enhanced drought tolerance by reducing water loss, increasing survival rate, and elevating stomatal conductance. These transgenic plants also showed higher leaf gas exchange, elevated abscisic acid (ABA) levels, and increased antioxidant and proline accumulation under drought stress [42]. Similarly, alfalfa expressing heterogeneous Osa-miR156bc Osa-miR156bcOE showed enhanced drought tolerance, as evidenced by greener leaf color and higher relative water content [43]. Additionally, miR156dOE in alfalfa enhanced salinity tolerance, manifested by increased biomass and shoot branching under salinity, along with higher total Kjeldahl nitrogen content under severe salinity stress [44]. Expression of Osa-miR156bc in alfalfa also conferred an improvement in salt tolerance, manifested by more branches and leaves, reduced leaf damage, and decreased Na^+^ content, malondialdehyde (MDA), and hydrogen peroxide (H_2_O_2_) levels [43]. Furthermore, alfalfa overexpressing miR156d exhibited enhanced heat stress tolerance, characterized by elevated non-enzymatic antioxidant content, water potential, anthocyanin content, and chlorophyll abundance [45]. Overexpression of miR156 enhanced Cd tolerance in Arabidopsis, promoting longer roots, increased biomass, higher chlorophyll content, and improved antioxidant activity, while miR156 silencing increased Cd sensitivity, leading to stunted growth and elevated oxidative stress [46]. This protective effect was attributed to reduced reactive oxygen species (ROS) accumulation and the upregulation of Cd transporters [46]. Similarly, in rice, miR156 overexpression reduced Cd uptake and ROS levels, enhancing Cd tolerance, whereas knockout plants accumulated higher Cd and ROS, increasing sensitivity [47]. In wheat and safflower, miR156 expression was downregulated in leaves but upregulated in roots, indicating a potential role in root-mediated Cd detoxification [48,49]. While the involvement of miR156 in response to various stress conditions has been investigated in many plant species, no studies have been published detailing its role in Al stress.

miRNAs are crucial for regulating plant physiology and development by controlling the expression of target genes. To fully understand miRNA function, loss of function (silencing) mutants are needed. Various approaches have been developed to downregulate miRNAs, including artificial miRNA (amiRNA) [50], anti-microRNA oligonucleotides (AMOs) [51], transcription-activator-like effector nucleases (TALENs), clustered regularly interspaced short palindromic repeats/CRISPR-associated nuclease 9 (CRISPR/Cas9) [52], target mimicry (TM) [39], and short tandem target mimic (STTM) [53]. Among these, short tandem target mimic (STTM) has emerged as the most effective strategy due to its stability, effectiveness in decoying miRNA [54], and ability to silence all members of a miRNA family simultaneously [55]. The mechanism of STTM involves the use of RNA molecules that mimic the natural target sites of miRNAs but are engineered to be more stable and resistant to degradation throughout plant development. This mimicry effectively sequesters miRNAs, preventing them from binding to their natural targets, thereby upregulating the target genes [53,55]. This method is effective in inducing specific developmental changes and is compatible with various miRNAs across different plant species [56]. For example, STTM has been used to improve drought tolerance in rice and disease resistance in tomatoes by targeting miRNAs involved in these pathways [57,58]. This approach offers significant potential for improving crop resilience and productivity under various environmental stresses [57,58].

In the current study, we investigated the expression profile of MsmiR156 in roots of WT alfalfa plants exposed to Al stress. We then used STTM technology to create genetically modified alfalfa plants with silenced miR156, tested the effects on the expression of target *SPL* genes, and analyzed the impact of silencing miR156 on alfalfa’s traits. We then conducted a comparative study to investigate the response of MsSTTM156 and MsmiR156 overexpression genotypes to Al stress. This study establishes a regulatory link between MsmiR156 and alfalfa’s response to Al stress.

## 2. Results

### 2.1. miR156 Is a Negative Regulator of Al Response in Alfalfa

The severity of Al toxicity in plants is influenced by multiple factors, including Al concentration, exposure duration, developmental stage, growth conditions, and plant species [21]. Previous research showed that 100 µM Al was optimal for evaluating Al-induced stress and its impact on plant growth in *Medicago sativa* [59], and thus this concentration was used in our study. To shed light on the role of miR156 in the regulation of Al response in alfalfa, transcript analysis was carried out on roots of wild-type (WT) plants subjected to 100 µM Al treatment. The analysis revealed significant miR156 silencing after 8 h of Al exposure compared to the control condition (Figure 1), suggesting a negative regulatory role for MsmiR156 in alfalfa’s response to Al stress.

### 2.2. Silencing of MsmiR156d Alters Expression of SPL Genes

To evaluate the efficiency of *miR156* silencing in alfalfa, we utilized STTM technology [55] to generate *miR156* knockdown alfalfa plants (MsSTTM156). This was achieved by doubling the number of tandem repeats and tripling the 48 nt spacer, creating a longer construct expected to enhance the suppression efficacy of *miR156* (Figure 2a,b). Using RT-qPCR analysis, we assessed the transcript levels of *miR156* and its putative target *SPL* genes in four independent MsSTTM156 alfalfa plants compared to WT (Figure 2b–l). The expression of mature *MsmiR*156 was reduced by 136% in genotype MsSTTM156-15, 45% in MsSTTM156-11, and 20% in MsSTTM156-12 relative to WT (Figure 2b). As miR156 was shown to downregulate at least 11 *SPL* genes in alfalfa [60], *MsmiR156* silencing influenced the expression of some of these genes in MsSTTM156 plants. Notably, six *MsSPL* genes, namely *MsSPL13*, *MsSPL13a*, *MsSPL6, MsSPL7a*, *MsSPL3*, and *MsSPL4* showed increased transcript levels in one-month-old leaves of MsSTTM156 plants (Figure 2c–h). However, there was some variability between *MsmiR156* downregulation and *MsSPL* expression in MsSTTM156 genotypes. For instance, while both *MsSPL2a* and *MsSPL9* increased in MsSTTM156-15, they decreased in MsSTTM156-11 and MsSTTM156-12 (Figure 2i,j). Conversely, *MsSPL12* showed an increase in MsSTTM156-12 and MsSTTM156-11 but a decrease in MsSTTM156-15 (Figure 2k). Similarly, *MsSPL3* showed increased expression in *MsSTTM156-11* and *MsSTTM156-15*, yet decreased in *MsSTTM156-12* (Figure 2g). Despite the substantial decrease in *miR156* levels, the relative transcript level of *MsSPL8* was reduced in all MsSTTM156 genotypes (Figure 2l). This variability suggests the possibility of alternative miR156 regulatory pathways or feedback mechanisms similar to the SPL15-miR156 regulatory loop identified in Arabidopsis [61].

### 2.3. Morphological Characterization of MsSTTM156 Transgenic Alfalfa

Prior to using MsSTTM156 plants in investigating their response to Al, we set out to determine the effects of miR156 silencing on alfalfa traits under non-stress conditions. To facilitate this analysis, both MsSTTM156 and WT control plants were propagated using vegetative cuttings for use in characterization. Prior to soil transplantation of plants at four weeks post-cutting, an initial phenotypic analysis was conducted and revealed a significant decline in root regeneration among the MsSTTM156 genotypes (Figure 3a). Specifically, root generation rates declined by 34–72% in MsSTTM156-11, MsSTTM156-12, and MsSTTM156-15 compared to WT plants (Figure 3a,b). Furthermore, root branching exhibited significant declines, ranging from 77 to 86%, across the MsSTTM156 genotypes relative to WT (Figure 3a,c). Root length was also substantially shortened, with reductions of 65%, 80%, and 82% observed in MsSTTM156-11, MsSTTM156-12, and MsSTTM156-15, respectively, compared to WT (Figure 3a,d).

These trends persisted over time. At one-month post-propagation, MsSTTM156 plants showed 43–78% shorter roots than WT (Figure 3d). By four months, root length lessened in severity for MsSTTM156-12 (11%) but remained significant in MsSTTM156-11 (41%) and MsSTTM156-15 (63%) relative to WT (Figure 3d). Similarly, root fresh weight and dry weight decreased significantly in a genotype-dependent manner, with the greatest reductions in MsSTTM156-15 compared to WT (Figure 3e,f). These observations emphasize the differential impact of miR156 silencing on root morphology and biomass across the MsSTTM156 genotypes.

Despite changes in root growth, the plant height of one-month-old MsSTTM156 plants showed no change compared to WT (Figure 4a,c). By the fourth month, however, plant height was 23–31% greater in MsSTTM156-12 and MsSTTM156-11 but 7% lower in MsSTTM156-15 relative to WT (Figure 4b,c). Furthermore, internode length showed no change in MsSTTM156 genotypes at one month (Figure 4d) but was 30–41% higher at four months (Figure 4d). Likewise, stem width was not different from WT at one-month (Figure 4e) but increased by 10–40% by the fourth month-old plants compared to WT (Figure 4e). In contrast, shoot branching showed a marked decline, with reductions of 44–67% in MsSTTM156-plants relative to WT at one month (Figure 4f). This trend deepened by the fourth month, with shoot branching declining significantly (51–73% lower) relative to WT (Figure 4f). This reduction in shoot branching had a significant impact on biomass, as evidenced by a 27–60% decline in shoot fresh weight (FW) (Figure 4g) and a 9–60% decrease in shoot DW observed in four-month-old MsSTTM156 plants compared to WT (Figure 4h). The MsSTTM156-12 genotype, which had the lowest reduction in miRNA156 expression levels, showed no significant difference in either FW or DW, suggesting a dose-dependent response to miR156 silencing in modulating both plant architecture and biomass accumulation.

MsSTTM156 plants showed distinct alterations in leaf morphology as manifested by two contrasting trends. Specifically, leaf area was reduced by 38% and 43% in genotypes MsSTTM156-12 and MsSTTM156-11, respectively, whereas MsSTTM156-15 displayed a 44% increase compared to WT (Figure 4i), suggesting a miR156 dose-dependency for this phenotype. Additionally, flowering time was impacted in MsSTTM156 plants, as MsSTTM156-12, and MsSTTM156-11 plants-initiated flowering about 17–19 days prior to WT (Appendix A). This early flowering in MsSTT156-12 and MsSTTM156-11 plants was associated with a 20 to 45% reduction in MsmiR156 transcript levels (Figure 2b). Conversely, the MsSTTM156-15 genotype, which exhibited the most pronounced reduction in MsmiR156 transcript levels (136%), failed to flower even after 157 days (Appendix A). These observed phenotypic changes are associated with altered miR156 expression levels, sometimes in a dose-dependent manner (e.g., leaf size and flowering time), underscoring the critical role of miR156 in modulating growth and development in alfalfa.

### 2.4. MsmiR156 Negatively Regulates Al Response by Altering Root Architecture

Consistent with prior studies reporting reduced *miR156* expression under Al stress across various plant species [62,63,64,65,66,67,68,69], our study further validated this finding, showing a significant reduction in *MsmiR156* transcript levels in alfalfa roots after exposure to 100 µM Al for 8 h (Figure 1). Based on this observation, we proceeded to generate MsmiR156 silencing plants (Figure 2, Figure 3 and Figure 4) to further investigate the role of MsmiR156 in regulating response to Al stress. Forty-nine-day-old MsmiR156 silencing (MsSTTM156-12, MsSTTM156-11, and MsSTTM156-15), overexpression (A11a, A17, and A8), and WT control plants (Figure 5) were exposed to 100 µM Al for 14 days. Under Al stress, MsSTTM156 plants revealed root length reductions of 11-42% in MsSTTM156-11, MsSTTM156-12, and MsSTTM156-15, compared to WT. In contrast, MsmiR156OE plants showed more noticeable reductions in root length, ranging from 49 to 64% in A11a, A8, and A17, respectively, relative to WT (Figure 5 and Figure 6a). When comparing Al-treated plants to their untreated controls, MsSTTM156-12 had a 37% reduction in root length, while MsSTTM156-11 and MsSTTM156-15 exhibited root length increases of 38% and 44%, respectively, compared to WT, which had a 22% reduction (Figure 5 and Figure 6a). Despite MsSTTM156-12 showing reduced root length, its roots remained longer than those of all MsmiR156OE genotypes, which experienced substantial reductions of 57%, 60%, and 64% in A8, A17, and A11a, respectively, compared to WT, which had a 22% reduction (Figure 5 and Figure 6a). Not only were the roots of MsmiR156OE and MsSTTM156 plants shortened under Al stress, but their color also changed to darker brown (Figure 5). Additionally, in terms of root fresh weight (FW) under Al stress, MsSTTM156 genotypes showed reductions of 18% in MsSTTM156-12 and 24% in MsSTTM156-15, while MsSTTM156-11 showed a 31% increase. Conversely, all MsmiR156OE genotypes displayed significant declines in root FW, with A11a, A17, and A8 showing decreases of 48%, 83%, and 91%, respectively, compared to WT (Figure 6b). When compared to their non-Al controls, MsSTTM156-12 suffered a 33% reduction in root FW, whereas MsSTTM156-11 and MsSTTM156-15 showed increases of 132% and 55%, respectively, compared to WT, which showed an 18% reduction (Figure 6b). By contrast, all MsmiR156OE genotypes had severe reductions in root FW, with A11a, A17, and A8 showing decreases of 67%, 92%, and 94%, respectively, relative to WT, which had only a 18% reduction (Figure 6b). Regarding root dry weight (DW), under Al stress, MsSTTM156-12 displayed a 16% reduction, while MsSTTM156-11 and MsSTTM156-15 exhibited increases of 77% and 12%, respectively, compared to WT. On the other hand, all MsmiR156OE plants showed significant decreases in root DW, with reductions of 41%, 71%, and 82% in A11a, A17, and A8, respectively, relative to WT (Figure 6c). When compared to their untreated counterparts, MsSTTM156-12 experienced a 17% reduction in root DW, whereas MsSTTM156-15 and MsSTTM156-11 exhibited increases of 16% and 70%, respectively, relative to WT, which had a 7% reduction (Figure 6c). By contrast, MsmiR156OE plants exhibited drastic reductions in root DW, with A11a, A17, and A8 showing decreases of 67%, 87%, and 92%, respectively, compared to WT, which showed a slight reduction of 7% under Al stress relative to no Al (Figure 6c). Collectively, these results suggest that MsmiR156OE negatively regulates root growth under Al stress in alfalfa, whereas silencing MsmiR156 results in minimal effects relative to WT, except for enhanced root DW in the MsSTTM156-11 genotype, which has a moderately reduced expression level of *MsmiR156*. Screening of more miR156 silencing genotypes would allow for determining whether certain miR156 expression levels could have more pronounced or desirable effects on plant traits under Al stress.

### 2.5. MsmiR156 Regulates Above-Ground Traits Under Al Stress

Given that MsmiR156 negatively impacts Al tolerance by impairing root growth, we further examined its effects on above-ground traits under Al stress. Phenotypic analysis revealed that under Al conditions, MsSTTM156-12 and MsSTTM156-15 had an 8% increase in plant height, while MsSTTM156-15 showed a 25% reduction compared to WT (Figure 5 and Figure 7a). Conversely, all MsmiR156 OE plants had significant plant height reductions of up to 56% compared to WT (Figure 5 and Figure 7a). Relative to their untreated counterparts, MsSTTM156-12 and MsSTTM156-11 displayed plant height increases of 8% and 47%, respectively, compared to a 3% decrease observed in WT (Figure 5 and Figure 7a). Although MsSTTM156-15 showed a slight 4% decrease in height, MsmiR156OE plants exhibited significant reductions of 32%, 37%, and 41% in A17, A11a, and A8, respectively, relative to WT (Figure 5 and Figure 7a). In terms of the effect of Al stress on stem diameter, MsSTTM156 genotypes showed noticeable increases of 59%, 61%, and 66% in MsSTTM156-12, MsSTTM156-15, and MsSTTM156-11, respectively, compared to WT (Figure 7b). In contrast, all MsmiR156OE genotypes had decreased stem diameters, with reductions of 39%, 47%, and 49% in A11a, A8, and A17, respectively, relative to WT (Figure 7b). Within the same genotype, MsSTTM156 plants showed increases of 4%, 25%, and 72% in stem diameter in MsSTTM156-11, MsSTTM156-12, and MsSTTM156-15, respectively, while MsmiR156OE plants showed declines of 37%, 43%, and 61% in A11a, A8, and A17, compared to a 4% reduction in WT (Figure 7b). Regarding shoot branching, there was a significant increase in MsSTTM156-12, while there was either no change or a reduction in MsSTTM156-11, MsSTTM156-15, and all MsmiR156OE plants compared to WT under Al stress (Figure 7c). Compared to their untreated counterparts, WT displayed a 29% decrease in shoot branching, whereas MsSTTM156-11 and MsSTTM156-12 showed notable increases of 30% and 130%, respectively (Figure 7c). Notably, no changes in shoot branching were observed in MsSTTM156-15, whereas reductions were evident in MsmiR156OE plants, ranging from 46% to 68% in A8, A11a, and A17, relative to WT under Al compared to no Al (Figure 7c). Furthermore, shoot fresh weight (FW) analysis of plants under Al stress revealed no significant change in MsSTTM156-12 and MsSTTM156-11, but MsSTTM156-15 had a reduction of 63% compared to WT. Conversely, all MsmiR156OE plants showed reductions of 79%, 82%, and 96% in A17, A11a, and A8, respectively, relative to WT (Figure 7d). Compared to their untreated counterparts, while MsSTTM156-12 had a reduction of 5% in shoot FW, MsSTTM156-15 and MsSTTM156-11 had reductions of 20% and 99%, respectively, compared to WT, which had a 17% reduction (Figure 7d). Similarly, all MsmiR156OE plants revealed sharp declines in shoot FW, with reductions of 63%, 74%, and 89% in A17, A11a, and A8, respectively, relative to a 17% reduction in WT (Figure 7d). Moreover, under Al stress, both MsSTTM156 and MsmiR156OE genotypes had significant reductions in shoot dry weight (DW) compared to WT (Figure 7e). However, compared to their untreated counterparts, MsSTTM156 plants consistently showed increases in shoot DW, with increases of 18%, 52%, and 209% in MsSTTM156-12, MsSTTM156-15, and MsSTTM156-11, respectively, compared to WT (Figure 7e). In contrast, MsmiR156OE plants showed substantial decreases in shoot DW, with A11a, A17, and A8 showing declines of 61%, 77%, and 84%, respectively, compared to WT, which had only a 6% reduction (Figure 7e). These results suggest that MsmiR156OE negatively regulates alfalfa above-ground traits, whereas MsmiR156 silencing enhanced stem diameter but had no impact on plant height and shoot FW under Al stress. MsSTTM156-12 enhanced shoot branching within the same genotype, suggesting a dose-dependent manner for this phenotype. These findings highlight the critical role of miR156 in regulating above-ground plant traits under Al stress.

### 2.6. MsmiR156 Modulates Physiological Changes in Response to Al Stress

Al stress is a well documented abiotic challenge that disrupts water uptake [70,71] and significantly reduces chlorophyll concentration [19]. Thus, the current study examined the effects of Al stress on relative water content (RWC) and chlorophyll levels in alfalfa plants with altered MsmiR156 expression, namely MsSTTM156 and MsmiR156OE. Despite Al stress having no notable effect on RWC in MsSTTM156 genotypes, it caused reductions ranging from 33% to 43% in MsmiR156OE genotypes compared to WT (Figure 8a). When compared to untreated controls, MsSTTM156-15 had a 27% decline in RWC, though this decline remained statistically comparable to WT and the other two silenced genotypes, MsSTTM156-12 and MsSTTM156-11 (Figure 8a). On the other hand, RWC reductions in MsmiR156OE plants were significantly higher, ranging from 22% to 30%, compared to the 6% reduction observed in WT (Figure 8a). Chlorophyll-A content largely mirrored RWC trends, except that chlorophyll content of MsSTTM156-15 did not exceed WT in the absence of Al stress. In MsSTTM156 genotypes, chlorophyll-A levels remained unaffected by Al stress, while significant reductions of 48% to 67% were recorded in MsmiR156OE plants compared to WT under Al stress (Figure 8b). Compared to untreated counterparts, relative to WT, MsSTTM156-11 genotype had a reduction of 13% in chlorophyll-A, but MsmiR156OE had more severe reductions of 56%, 57%, and 71% in A17, A11a, and A8, respectively (Figure 8b). Chlorophyll-B revealed contrasting trends. While there were increases of 41% and 49% observed in MsSTTM156-12 and MsSTTM156-15, respectively, dramatic reductions of 57%, 65%, and 68% were observed in A11a, A17, and A8, respectively, compared to WT under Al stress (Figure 8c). Within the same genotype, comparisons revealed distinct trends: chlorophyll-B content rose by 80% and 136% in MsSTTM156-15 and MsSTTM156-11, respectively, but declined markedly by 67% to 85% in MsmiR156OE genotypes relative to WT, which had a reduction of only 9% (Figure 8c). Total chlorophyll content mirrored these findings. Increases of 19% and 22% were observed in MsSTTM156-12 and MsSTTM156-15, respectively, compared to WT under Al stress (Figure 8d). In contrast, MsmiR156OE genotypes suffered severe reductions of 49% to 66% relative to WT (Figure 8d). Compared to untreated controls, total chlorophyll content increased by 10% and 30% in MsSTTM156-11 and MsSTTM156-15, respectively, relative to WT. Conversely, MsmiR156OE plants showed significant reductions, ranging from 64% to 77%, compared to a 4% reduction observed in WT (Figure 8d). These findings suggest that miR156 plays a critical role in modulating both RWC and chlorophyll concentration in alfalfa under Al stress.

### 2.7. MsmiR156 Regulates IAA Transport Genes Under Al Stress

Al toxicity is detrimental to plant development, as evidenced primarily by its inhibition of root growth [72]. This inhibition is closely associated with disrupted distribution and accumulation of indole acetic acid (IAA) in the root tips, which is critical for root elongation and development [21,73]. To shed light on this mechanism in alfalfa, we investigated the transcript levels of the IAA transport-related genes, *MsAUX1* and *MsPIN2*, in root tips of 49-day-old MsSTTM156, MsmiR156OE, and WT exposed to 100 µM Al for 24 h. Following Al treatment, the transcript levels of *MsAUX1* were reduced by 2%, 38%, and 29% in MsSTTM156-12, MsSTTM156-11, and MsSTTM156-15, respectively (Figure 9a), whereas *MsAUX1* levels in MsmiR156OE roots were enhanced by between 31% and 52%, relative to WT (Figure 9a). Similarly, *MsPIN2* transcript levels showed reductions ranging from 27% in MsSTTM156-11 to 39% in MsSTTM156-15 relative to WT (Figure 9b). In MsmiR156OE plants, on the other hand, *MsPIN2* was elevated by 34%, 42%, and 46% in A11a, A8, and A17, respectively, relative to WT (Figure 9b). Together, these results shed light on a potential root inhibition mechanism, in which MsmiR156 regulates IAA transport-related genes to either increase or decrease the accumulation of IAA in the roots of alfalfa under Al stress.

### 2.8. Expression of MsPG Genes Is Affected by miR156 Under Al Stress

The elevated expression of polygalacturonase genes, *MsPG1* and *MsPG4,* in alfalfa root tips led to decreased pectin levels within the cell walls, thus increasing porosity and flexibility in the cellulose framework, ultimately enhancing tolerance to Al stress [74,75]. To investigate whether MsmiR156 plays a role in determining cell extensibility, we measured transcript abundance of the cell wall-associated genes, *MsPG1* and *MsPG4***,** in roots of 49-day-old MsSTTM156 and MsmiR156OE relative to WT alfalfa roots subjected to 100 µM Al for 24 h. Post-Al treatment, *MsPG1* transcript levels were higher in MsSTTM156-15 and MsSTTM156-11 but remained largely unchanged in MsSTTM156-12 relative to WT (Figure 9c). In contrast, *MsPG1* was downregulated in all MsmiR156OE genotypes under Al compared to both the untreated counterpart and WT (Figure 9c). Likewise, analysis of *MsPG4* transcript levels revealed an increase of 1.31-fold in MsSTTM156-11 but no significant change in MsSTTM156-12 and MsSTTM156-15 relative to WT (Figure 9d). MsmiR156OE genotypes, on the other hand, showed a significant reduction in *MsPG4* transcript levels compared to WT under Al stress (Figure 9d). These results highlight the intricate regulatory role of MsmiR156 in modulating the expression of key cell-wall-related genes, thereby influencing cell extensibility and ultimately contributing to enhanced Al tolerance in alfalfa.

## 3. Discussion

Al toxicity is one of the most critical factors that limit crop production in acidic soils worldwide. The highly conserved miR156 regulates various plant characteristics by silencing the expression of genes encoding transcription factors from the SPL family [76]. In the present study, we used alfalfa plants with silenced or enhanced *MsmiR156* expression to investigate the regulatory role of MsmiR156 in Al stress in this plant. The *MsmiR156*-overexpressing (*MsmiR156OE*) plants used here were initially characterized in a previous study under control conditions [35]. The data obtained from these plants were compared with findings from plants with suppressed *miR156* expression, as shown in both this and the earlier study, providing valuable insights into the contrasting effects of *miR156* regulation on various agronomic traits in alfalfa. We demonstrated that MsmiR156 negatively regulates alfalfa’s response to Al stress. Earlier findings in alfalfa showed that miR156 governs key traits such as root development, nodulation, plant architecture, forage yield, flowering time, and response to abiotic stress [35,42,45,60]. Here, we specifically showed that MsmiR156 is downregulated in the roots under Al stress (Figure 1). This downregulation aligns with patterns reported in several other plant species [62,64,65,66,67,68], further suggesting a conserved regulatory role of MsmiR156 in response to Al-induced stress in plants. However, it presently remains unclear which downstream miR156-targeted SPLs mediate the effect of miR156 on Al response.

### 3.1. miR156 Silencing Modulates SPL Genes in Transgenic Alfalfa

Using the highly effective STTM technology [55], we successfully developed transgenic alfalfa plants with reduced *MsmiR156* expression (Figure 2b), leading to the upregulation of target *MsSPL* genes (Figure 2c–k). This finding is in accordance with previous research in Arabidopsis, where STTM156/157 plants showed a reduction in miR156 expression and an increase in *SPL* transcripts, including *SPL3*, *SPL4*, *SPL5*, *SPL15*, and *SPL9* [56]. However, our data revealed a lack of consistent correlation between MsmiR156 silencing and the level of MsSPL upregulation across transgenic plants. For instance, *MsSPL3* was downregulated in MsSTTM156-12 plants while upregulated in MsSTTM156-11 and MsSTTM156-15 plants (Figure 2g), presumably due to the miR156 dose-dependent effects. This finding mirrors previous observations in MsmiR156OE alfalfa, where *MsSPL* expression did not always inversely correlate with *MsmiR156* levels [35]. This inconsistency of MsmiR156-dependent *MsSPL* transcript levels suggests the involvement of additional regulatory mechanisms beyond the direct MsmiR156-SPL interaction, as previously noted in other species [61]. It is possible that other unidentified factors or post-transcriptional mechanisms could be modulating this interaction. Understanding these complex regulatory networks is essential for advancing our knowledge of the *miR156*-mediated stress response. Future studies should aim to identify these unknown factors and elucidate their role in fine-tuning gene expression. Such investigations could yield deeper insights into the broader regulatory networks governing the miR156-SPL pathway, leading to more precise genetic strategies for crop improvement.

### 3.2. miR156 Silencing Reduces Root Growth in Alfalfa

Morphological analysis of MsmiR156-silenced plants highlights the critical role of miR156 in regulating root architecture, as evidenced by the impairments in root regeneration (Figure 3b), root branching (Figure 3c), root length (Figure 3d), and root biomass (Figure 3e,f) observed in MsSTTM156 plants. These results align closely with prior findings in Arabidopsis, where reduced expression of miR156 (MIM156) led to shorter primary roots and fewer lateral roots due to elevated expression of *SPL* genes, such as *SPL*3 and *SPL*9 [77]. Similarly, MsSTTM156 plants exhibited increased expression of *MsSPL*3 and *MsSPL*9 (Figure 2g,i), further supporting the inhibitory role of these genes in root growth regulation. Conversely, elevated expression of miR156 (MsmiR156OE) enhanced root regeneration and biomass accumulation, consistent with previous observations in alfalfa [35], maize [78,79], *Lotus japonicus* [80], and red clover [81], suggesting a broad and positive influence of miR156 on plant root growth traits.

### 3.3. miR156 Affects Leaf Pigments in Alfalfa

Our findings revealed that the leaves of MsSTTM156 plants displayed a notably darker green color compared to WT (Appendix A), suggesting alteration in pigment biogenesis or degradation pathways. Interestingly, despite this darker phenotype, chlorophyll measurements indicated either no significant difference or slightly reduced levels in MsSTTM156 plants relative to WT (Figure 8a–c). This discrepancy suggests a complex role for miR156 in regulating leaf pigment metabolism, potentially affecting chlorophylls, carotenoids, and anthocyanins. Previous research in Arabidopsis revealed that reduced miR156 expression elevated SPL transcription factors involved in specialized metabolism [56]. Thus it is possible that certain SPLs may influence chlorophyll degradation through genes such as NON-YELLOW COLORING1 (*NYC1*) and STAY-GREEN (*SGR*). Future research should clarify how miR156-SPL regulatory networks influence pigment dynamics and photosynthetic performance in alfalfa.

### 3.4. miR156 Silencing Alters Flowering in Alfalfa

We next investigated the impact of knocking down MsmiR156 on flowering time in MsSTTM156 plants, which showed early flowering compared to WT (Appendix A).This aligns with findings in Arabidopsis, where silencing miR156 similarly induced early flowering [56]. This early flowering phenotype likely resulted from increased expression of genes encoding floral activators or integrators, including *SOC1*, *AGL24*, and *LFY*, coupled with reduced expression of genes encoding floral repressors such as *AP2* and *FLC* [56]. Interestingly, severe silencing of miR156 in the MsSTTM156-15 genotype delayed flowering, mirroring the delayed flowering phenotype observed in MsmiR156OE plants, where miR156 was overexpressed by approximately 840-fold–1200-fold [35]. The significant delay in flowering in MsmiR156OE plants is partially explained by the downregulation of the *AP3* gene, a crucial regulator of flowering time [35]. This delayed flowering in MsmiR156OE plants is partially explained by the downregulation of the *AP3* gene, a crucial regulator of flowering time in these plants [35]. However, the mechanism behind the similar flowering delay caused by both silencing and excessive overexpression of miR156 remains unclear and requires further investigation into possible feedback loop mechanisms within the flowering regulatory network under the two contrasting conditions. We hypothesize that the regulation of flowering by the two extremes of miR156 expression may be due to similar effects on the expression of floral repressors. For example, silencing *miR156/157* and *miR172* in Arabidopsis upregulates *TEM1* and downregulates *SM2*, while *miR156/157* and *miR172* inversely regulate each other to modulate flowering timing [56]. Despite their opposing regulatory roles in flowering in Arabidopsis [82], miR156/157 and miR172 play predominant roles in controlling flowering time by regulating flowering genes through interactions with SPL targets [56]. This suggests that SPLs, in coordination with *miR172*, may mediate the delayed flowering phenotypes in both *MsmiR156OE* and *MsSTTM156-15* plants. However, in general, silencing of *miR156* induces early flowering, whereas its overexpression delays flowering [35].

### 3.5. miR156 Role in Root Growth Under Al Stress

Al toxicity is a major factor limiting root growth in alfalfa [14,59]. Our findings showed that miR156OE exacerbated root growth inhibition relative to WT under Al stress (Figure 6a). These findings differ from those reported in Arabidopsis, where miR156OE improved cadmium stress tolerance by promoting root elongation, while its silencing reduced root growth [46]. Additionally, our results revealed up to a 94% decrease in root FW and 92% in DW in MsmiR156OE plants, further highlighting the detrimental effects of miR156OE under Al stress (Figure 6b,c). Conversely, MsSTTM156-11 displayed a 70% increase in dry weight within the same genotype under Al stress compared to no Al, further supporting the potential benefits of miR156 silencing (Figure 6b,c). Our findings of reduced root length and biomass under Al stress align with an earlier alfalfa study, where exposure to 100 μM AlCl_3_ led to a reduction in the root length by 17–21%, alongside a decrease in root FW [59]. Similarly, in alfalfa under Al stress, shoot and root biomass were significantly lower than those of the control [18]. The root growth impairments observed in this study are likely due to Al interference with cell division and elongation in the root apical meristem, resulting in stunted, brownish roots with diminished fine branching and root hair formation, as reported in previous studies on Al toxicity [21,72,83,84,85]. Another contributing factor may be the reduction of indole-3-acetic acid (IAA) levels. Previous research has shown that Al toxicity in alfalfa disrupts the transport and accumulation of IAA in apical buds and root tips, leading to reduced IAA levels. This reduction contributes to cellular disorganization, deformed cell shapes, a shortened meristematic zone, and ultimately inhibits root growth [59], suggesting a multifaceted miR156-based regulation of root growth in response to Al stress in alfalfa.

### 3.6. miR156 Regulates Root Browning Under Al Stress in Alfalfa

Al toxicity induces root shortening, thickening, and browning [86]. Our findings revealed that *miR156* modulates root color under Al stress, as *MsSTTM156* and *MsmiR156OE* plants developed darker brown roots (Figure 5, Appendix A). This observation aligns with prior studies on *Sophora davidii* seedlings, where increasing Al concentrations and acidity progressively deepened root pigmentation, indicating strong root sensitivity to Al toxicity [86]. The browning of roots under Al stress is primarily attributed to the accumulation and oxidation of phenolic compounds, particularly catechins, as reported in *Pinus massoniana* [87]. Al toxicity upregulates genes involved in the phenylpropanoid and flavonoid biosynthesis pathways, leading to enhanced catechin production. While these compounds are secreted as root exudates to chelate Al ions, their oxidation contributes to root browning [87]. Furthermore, Al-induced oxidative stress generates reactive oxygen species (ROS), triggering lipid peroxidation and protein oxidation, which further compromise cellular integrity and intensify root browning [88]. This suggests that the change in root color to darker brown in this study may be due to a combination of phenolic compound accumulation, oxidative stress, and metabolic disruptions induced by Al toxicity, potentially regulated by miR156.

### 3.7. MsmiR156 Regulates Shoot Architecture and Biomass Traits Under Al Stress

Al stress not only impairs root growth but also significantly reduces overall plant biomass [89]. We found that MsmiR156OE reduced plant height, whereas silencing this gene (MsSTTM156) resulted in no evident change in plant height under Al stress (Figure 7a). The reduction of the plant height aligns with two previous studies in alfalfa that showed Al treatment had a negative impact on this trait [14,90]. Additionally, we showed that miR156OE caused significant reductions in stem basal diameter in MsmiR156OE plants under Al (Figure 7b). Conversely, miR156 silencing enhanced stem basal diameter in MsSTTM156 plants under Al (Figure 7b), suggesting a complex role for miR156 in regulating stem diameter under this stress.

Silencing of MsmiR156 in MsSTTM156-12 genotype also enhanced shoot branching under Al stress, suggesting a dose-dependent effect of miR156 on this trait, whereas MsmiR156OE had a negative impact on this phenotype under Al stress (Figure 7c). This indicates that miR156 plays a critical role in regulating shoot architecture in response to Al stress. This complex regulation may extend beyond miR156, and may involve interactions with other key players such as the pectin acetylesterase12 gene (*MsPAE12*), which is highly expressed in alfalfa shoot apices and has been shown to regulate shoot branching [91]. Overexpression of *MsPAE12* increases the number of shoot branches, whereas its silencing causes reductions [91]. The mechanism underlying this regulation appears to be linked to auxin (indole-3-acetic acid, IAA) signaling, a central player in suppressing axillary bud outgrowth and modulating shoot branching [92], and that *MsPAE1*2 negatively regulates acetic acid (AA) content in the shoot apex and cell wall, thereby reducing IAA biosynthesis and promoting shoot branching [91]. It is possible that miR156 influences shoot branching through the MsPAE12-mediated IAA biosynthesis pathway. The reduced shoot branching observed in miR156OE plants may be linked to increased IAA accumulation in the shoot apex, suggesting a complex interplay between miR156 and the MsPAE12-mediated IAA biosynthesis pathway to regulate the shoot branching trait under Al stress.

Our findings highlight a distinct role for MsmiR156 in regulating shoot biomass under Al stress. While MsSTTM156-12 and MsSTTM156-11 genotypes showed no significant change in shoot fresh weight under Al stress relative to WT (Figure 7d), shoot dry weight was significantly reduced in both genotypes compared to WT (Figure 7e). Notably, under non-Al stress, both shoot fresh weight (Figure 7d) and dry weight (Figure 7e) were significantly lower in MsSTTM156-11 genotype relative to WT. This observation points to a potential dose-dependent effect of miR156 silencing on shoot biomass traits. By contrast, there were 63–89% reductions in shoot fresh weight in MsmiR156OE plants (Figure 7d), accompanied by 61–84% reductions in shoot dry weight relative to WT under Al compared to no Al (Figure 7e). These findings contrast with findings in *Arabidopsis* and rice studies, where *miR156*OE enhanced shoot biomass, while its suppression led to a significant reduction under cadmium stress [46,47]. These contrasting findings underscore the species- and stress-specific nature of miR156’s regulatory role, suggesting that miR156 may employ distinct mechanisms depending on the type of environmental stress and the plant species involved.

The reduced shoot biomass observed in MsmiR156OE alfalfa under Al stress aligns with previous findings across several species, including maize, wheat, sorghum, oat [93], *Nigella arvensis* [94], and *Zea mays* [95]. At the molecular level, our results indicate that the biomass reduction in MsmiR156OE plants is primarily driven by altered root architecture caused by disrupted auxin (IAA) transport. Specifically, overexpression of miR156 significantly upregulates the auxin transporter genes *MsAUX*1 and *MsPIN*2, disturbing auxin distribution patterns and consequently inhibiting root elongation. This impaired root growth negatively affects water and nutrient uptake, directly limiting shoot biomass accumulation. Additionally, Al toxicity exacerbates nutrient imbalances by disrupting the uptake and allocation of essential elements such as calcium, nitrogen, potassium, magnesium, phosphorus, manganese, iron, copper, and boron [96,97].These imbalances further disrupt critical cellular processes, including transmembrane potential maintenance and H+-ATPase activity, essential for optimal plant growth [98,99]. Thus, future research should focus on elucidating the precise molecular interactions between miR156, auxin transport pathways, and nutrient acquisition mechanisms to develop targeted strategies for enhancing alfalfa productivity under Al stress.

### 3.8. MsmiR156’s Influences on Water Dynamics and Chlorophyll Content Under Al Stress

Our findings highlight an essential role for MsmiR156 in modulating water uptake and chlorophyll content under Al stress. Silencing miR156 in MsSTTM156 genotypes preserved relative water content (RWC), whereas MsmiR156OE plants showed significant RWC reductions compared to WT under Al stress (Figure 8a). Prior research in alfalfa showed that miR156OE maintained RWC under drought stress [42]. The reduced RWC in MsmiR156OE plants under Al is likely due to impaired root elongation, swelling of root apices, and a reduction of root hair formation, as reported in previous studies [100,101,102]. Prolonged exposure to Al toxicity exacerbates these issues, further leading to impaired water uptake and ultimately reducing plant growth and productivity [103,104]. Consistent with these observations, similar disruptions have been reported in tomatoes under Al stress [105], suggesting that miR156OE impedes water uptake under Al, likely by inhibiting root elongation and restricting the root’s capacity to access water.

Chlorophyll content displayed a similarly genotype-dependent response under Al stress. While chlorophyll-A levels remained stable in MsSTTM156 genotypes, MsmiR156OE plants showed significant reductions relative to WT under Al stress (Figure 8b). Notably, chlorophyll-B and total chlorophyll levels increased in MsSTTM156-12 and MsSTTM156-15 genotypes but declined in MsmiR156OE plants under the same conditions (Figure 8c,d). These findings are consistent with prior studies reporting Al-induced chlorophyll decline across species, including barley [106], spinach [107], maize [108], rye (*Secale cereale*) [109], peanut (*Arachis hypogaea*) [110], cocoa (*Theobroma cacao*) [111], *Eucalyptus* [112], rice [113], highbush blueberry (*Vaccinium corymbosum*) [114], alfalfa (*Medicago sativa*) [18]. However, the reductions in chlorophyll content observed in MsmiR156OE plants under Al contrast with previous findings in Arabidopsis, where miR156OE improved chlorophyll content under cadmium stress [46], and in alfalfa under heat stress [45].

The decline in chlorophyll content in MsmiR156OE plants under Al stress may be attributed to the inhibition of key biosynthetic enzymes. Al^3+^ ions interfere with δ-aminolevulinic acid dehydratase (ALA-D), an enzyme essential for the formation of porphobilinogen, a precursor to chlorophyll. This inhibition occurs as Al^3+^ competes with Mg^2+^ for the enzyme’s active site, thereby impairing the synthesis of chlorophyll [115]. Additionally, Al stress reduces the uptake of Mg and Fe, critical cofactors for chlorophyll production, further exacerbating the deficiency [116]. Together, these disruptions explain the chlorophyll deficiency observed in MsmiR156OE plants under Al stress. These findings suggest multiple roles for MsmiR156 in stress adaptation, particularly in the interplay between water dynamics and chlorophyll content. Future investigations should examine how *MsmiR156* impacts chlorophyll biosynthesis and water uptake pathways, with an emphasis on interactions involving nutrient acquisition and hormonal regulation. Unraveling these mechanisms could offer novel strategies to enhance alfalfa’s resilience to Al toxicity.

### 3.9. MsmiR156 Regulates IAA Transport in the Roots Under Al Stress

Our investigation into the molecular mechanisms underlying MsmiR156’s role in root growth inhibition revealed its significant impact on the expression of IAA transport-related genes *MsAUX1* and *MsPIN2*. These genes are critical mediators of auxin transport and root development. Reduced expression of *MsAUX1* and *MsPIN2* in MsSTTM156 plants correlates with a decrease in Al-induced root growth inhibition. In contrast, MsmiR156OE led to the upregulation of these genes, amplifying the effects of Al toxicity on root development (Figure 9a,b). These results are in accordance with a previous finding that the upregulation of both *MsAUX1* and *MsPIN2* in root tips of alfalfa plants exposed to Al^3+^ enhanced polar auxin transport from the quiescent center of root tips to the upper root, which resulted in a higher IAA accumulation in the epidermis and cortex of the meristematic zone [59]. These findings suggest that MsmiR156 plays a critical role in the distribution of IAA, which is essential for Al stress response and developmental processes in alfalfa. Whereas fine tuning miR156 expression shows promise in improving Al stress tolerance in alfalfa, future research should investigate how miR156 interacts with other hormonal pathways during Al stress and whether it influences signaling molecules beyond IAA. Experiments tracking changes in auxin distribution at the cellular level, along with mutant analyses of other IAA-related genes, could offer a deeper understanding of this regulatory network.

### 3.10. miR156 Modulates Cell Wall Extensibility and Al Tolerance in Alfalfa Roots

Our results also revealed that MsmiR156 regulates directly or indirectly two genes linked to cell wall extensibility, *MsPG1* and *MsPG4,* as manifested by their upregulation and downregulation in MsSTTM156 and MsmiR156OE genotypes, respectively (Figure 9c,d). These results align with previous studies showing that overexpression of *MsPG1* and *MsPG4* increased root elongation and reduced Al accumulation in pectin, hemicellulose 1 (HC1), cell walls, and root apexes in alfalfa [74,75]. The reduction in Al content was attributed to lower levels of HC1 and fewer negatively charged functional groups (–OH and –COOH) in the cell wall, which typically bind Al [75]. Consequently, this decrease protected the microstructure of root apexes, increased the porosity and pore size of the lattice-like cell wall structure, and facilitated better root growth under Al stress [75]. Similarly, overexpression of *MsPG4* in alfalfa reduced both water-soluble pectin (WSP) and chelator-soluble pectin (CSP) in the cell wall and reduced Al accumulation, while silencing *MsPG4* led to increased Al accumulation and pectin content [74]. The overexpression of *MsPG1* and *MsPG4* not only decreased pectin polymer size but also enhanced the porosity and flexibility of the cell wall network, resulting in improved Al tolerance through reduced Al accumulation in root tips of alfalfa plants [74,75]. This suggests that downregulation of MsmiR156 could potentially enhance cell wall flexibility (reducing cell wall modification), thus contributing to a root architecture that may be more resilient to Al stress. To fully understand the potential of MsmiR156 modulation, future research should examine its interactions with other hormones and signaling pathways that influence cell wall structure and function.

## 4. Materials and Methods

### 4.1. Plant Materials and Growth Conditions

Transgenic alfalfa (*Medicago sativa*) plants expressing miR156 (miR156OE) were previously generated in our laboratory at Agriculture and Agri-Food Canada as described in Aung et al., 2015 [35]. Wild-type (WT) alfalfa clone N4.4.2 [117] was obtained from Dr. Daniel Brown (Agriculture and Agri-Food Canada) and was used in transformation experiments and for all other purposes of this work. Transgenic and WT alfalfa plants were cultivated and maintained in a greenhouse under controlled conditions at temperatures between 21 and 23 °C, with 16 h of light exposure (halogen lighting after 18:00 h), light intensity ranging from 380 to 450 W/m^2^, and relative humidity (RH) of 70%.

### 4.2. Generation of MsSTT156 Constructs and Alfalfa Transformation

To create the MsSTTM156 construct, STTM targets were initially designed based on the mature alfalfa miR156 (MsmiR156d) and the mRNA sequence complementary to the miR156 binding site of *MsSPL* transcripts [35]. A trinucleotide mismatch (bulge) was incorporated into the binding site of *SPLs* to prevent cleavage by mature miR156. The MsSTTM156 construct was designed, synthesized (as a fee for service by Bio Basic, Toronto, ON, Canada), and cloned using the Gateway cloning system (Thermo Fisher Scientific, Mississauga, ON, Canada). A 500 bp sequence was amplified from the synthesized sequence of MsSTTM156 using primers with *Att*B sites attached, STTM156-GWF and STTM156-GWR (Appendix A). The amplicon was cloned into the pDONR/Zeo entry vector and transferred to *E. coli* DH5α by electroporation. After PCR screening of *E. coli* transformants and confirmation by sequencing, the fragment from the construct was subcloned into the pEarley-Gate101 (Invitrogen, Carlsbad, California, USA) as the destination vector using LR clonase II enzyme mix reactions. Expression clones *pEG101:MsSTTM156* were verified by colony PCR using gene-specific forward and reverse primers (Appendix A) and validated by sequencing. The expression vectors were then transferred to *Agrobacterium tumefaciens* EHA105 by electroporation, and the transformed strain was used to transform alfalfa N4.4.2. Plant transformation was performed as described by [118]. The presence of the transgene in putative MsSTTM156 alfalfa genotypes was confirmed by PCR using gene-specific forward and reverse primers (Appendix A). qRT-PCR was then used to analyze transcript levels of *MsmiR156* and *MsSPL* genes using respective primers (Appendix A).

### 4.3. Vegetative Propagation of Alfalfa Plants by Stem Cuttings

Due to the obligatory outcrossing reproduction of alfalfa, plants were propagated by vegetative cuttings to ensure that the genetic background is maintained throughout the experiment. To synchronize growth and ensure all plant materials were at the same developmental stage before vegetative propagation by rooted stem cutting, alfalfa plants were cut back three times and allowed to grow for one month. The vegetative propagation was performed by making stem-cuttings of 3–4 segments, each containing 2–3 nodes, which were cut and inserted into moistened growing media (Pro-Mix^®^, Mycorrhizae^™^, Premier Horticulture Inc., Woodstock, ON, Canada) in a plastic tray. The tray was covered with propagation domes (Ontario Grower’s Supply, London, ON, Canada) and maintained in the greenhouse for four weeks to facilitate the root formation from the cut stem. Rooted cuttings were transferred to BX Mycorrhizae (PRO-MIX^®^, Smithers-Oasis North America, Kent, OH, USA) soil in 6” plastic pots. The plants were maintained under greenhouse conditions for further plant development and characterization.

### 4.4. Morphological Characterization of Alfalfa Plants

To characterize the phenotypic variations in MsSTTM156 alfalfa compared to WT control and miR156OE, plants were generated from stem cuttings, maintained in the greenhouse, and watered as required. The plants were characterized at four weeks post-stem-cutting and at one- and four-months post-transfer to soil. The characteristics assessed included root formation (regeneration), root length, root branches, plant height, internode length, number of main branches, leaf area, stem diameter, flowering time, fresh weight (FW), and dry weight (DW). The rooted stems were counted and subtracted from the total stem number of each tray per genotype. At least three to five trays were used for this experiment, with each tray containing 50 stem cuttings. The root length was measured using the primary and most extended root emerging at the base of the stem, the length between the neck to the tip, whereas those on the primary root branches were counted as lateral root branches. The most extended stem in each biological replicate was used to measure plant height and the internode length. The main shoot branches were defined as directly emerging from the soil. The stem width (diameter) in each replicate was measured from the base of the first centimetre of the main stem. Fresh weight (FW) of the tissue was recorded at harvest time, while dry weight (DW) was obtained after drying the tissue at 65 °C for five days. Leaf area was measured by photographing five leaves from each plant (biological replicate/genotype) using a Canon camera. The images were subsequently analyzed using Python-based tools(Version3.11.5). The custom Python pipeline (Appendix A) was specifically designed for image processing, employing the PlantCV library to accurately quantify or measure leaf area (leaf size).

### 4.5. Al Treatment

To investigate the regulatory role of miR156 in alfalfa’s response to Al stress, MsSTTM156, MsmiR156OE and WT plants were subjected to Al stress. For Al treatment, rooted stem cuttings were transferred to BX Mycorrhizae (PRO-MIX^®^, Smithers-Oasis North America, Kent, OH, USA) soil in 6” plastic pots for 35 days. The plants were then transferred to 3.2 L plastic buckets containing modified Hoagland solution [119] with an adjusted pH of 4.5 for acclimatization for 14 days. After acclimatization, the 49-day-old plants were treated with a 100 μM AlCl_3_·6H_2_O solution or no Al (0 μM Al) for 14 days. Fresh AlCl_3_·6H_2_O solutions were applied every three days for Al treatment.

### 4.6. Analysis of Relative Water and Chlorophyll Contents

To determine the effect of miR156 on the relative water content (RWC) of alfalfa plants exposed to Al stress, 0.5 g of fresh trifoliate leaves were initially cut and weighed (FW), immersed in deionized water in sealed glass jars, and incubated in the dark for 48 h at 4 °C. After removing water, excess moisture was blotted off, and leaves were reweighted to obtain the fresh turgid weight (FTW). Subsequently, the FTW samples were dried at 65 °C for five days and then weighed to determine DW. RWC was calculated using the following equation, as described in [120]:$$RWC=\left[\frac{FW-DW}{FTW-DW}\right]\times 100$$

To assess chlorophyll concentration, 0.15 g of fresh trifoliate leaves of 63-day-old MsSTTM156, Ms miR156OE, and WT plants were collected and immediately frozen in liquid nitrogen before chlorophyll extraction. The tissue samples were homogenized in 2 mL of a 1 M NH_4_OH: C_3_H_6_O mixture (1:9 *v*/*v*). The volume was then adjusted to 5 mL with 80% aqueous acetone, and the mixture was refrigerated for 2 h in the dark. After centrifugation for 20 min at 500× *g*, the supernatant was transferred to a fresh tube, and the volume was adjusted to 10 mL with 80% acetone. Using 80% acetone as a blank, the absorbances of five biological and four technical replicates were measured using a spectrometer (Smartspec™ Plus, Bio-Rad, Hercules, CA, USA). Absorbance was recorded at 663 nm for (chlorophyll-A) and 645 nm for (chlorophyll-B). The concentration of chlorophyll pigments in a solution was determined using the following equations as described in [121]:Chlorophyll-A (mg/mL) = 12.7 × A663 − 2.69 × A645Chlorophyll-B (mg/mL) = 22.9 × A645 − 4.68 × A663Total Chlorophyll (mg/mL) = Chlorophyll-A + Chlorophyll-B

Here, A663: Absorbance at 663 nm, corresponding to the maximum absorption of Chlorophyll-A, A645: Absorbance at 645 nm, corresponding to the maximum absorption of Chlorophyll-B, mg/mL: Milligrams of chlorophyll per milliliter of solvent.

### 4.7. RNA Extraction and Reverse Transcription-Quantitative PCR

Fresh leaves and root tips of alfalfa were collected, flash-frozen in liquid nitrogen, and stored at −80 °C until further use. Approximately 100 mg fresh weight was used to extract total RNA using the RNeasy Plant Mini Kit (Qiagen, Germantown, MD, USA, Cat # 74904) for leaf samples and the Total RNA Purification Kit (Norgen Biotek, Thorold, ON, Canada, Cat #25800) for root tips. The tissue was then homogenized using a PowerLyzer ^®^24 bench-bead-based homogenizer (Cat #13155) following the manufacturer’s instructions. DNA digestion was performed using a TURBO DNA-free kit (Invitrogen, Waltham, MA, USA, Cat # 2835993). Subsequently, cDNA was synthesized from 2 µg of total RNA using the SuperScript^®^IV Reverse Transcriptase (RT) (Invitrogen, Waltham, MA, USA, Cat # 18090010). For RT-qPCR analysis, the SsoFast™ EvaGreen^®^Supermix (Bio-Rad, Hercules, CA, USA, Cat # 6116-1725204) and gene-specific primers were utilized for RT-qPCR analysis. Each reaction consisted of 4 μL of cDNA template, 0.5 μL of forward and reverse gene-specific primers (10 μM each) (Appendix A), and 5 μL of SsoFast EvaGreen Supermix. The analysis was conducted with at least three biological and three technical replicates for each plant genotype. Data were analyzed using the CFX Maestro™ software ( version 3.1) (Bio-Rad), and relative gene expression ratios were calculated using the Vandesompele method [122]. All RT-qPCR data were normalized against the reference genes *CoA carboxylase 1* and *acetyl CoA carboxylase 2* [122]. The normalized relative gene expression was log-transformed at the base of 10 before conducting statistical analysis to normally distribute the data as described in [123]. Graphs were generated using GraphPad Prism 10.0 (www.graphpad.com).

### 4.8. Statistical Analysis

Statistical analyses were conducted using R (version R-4.2.3) tools. A Student’s *t*-test was employed to assess the significant differences between WT plants treated with 100 µM Al and those untreated (0 µM Al) under control conditions. A one-way ANOVA was used to compare gene expression of *SPL* genes and morphological changes among the MsSTTM156 genotypes and WT, while two-way ANOVAs assessed the impact of Al treatment across MsSTTM156, MsmiR156OE, and WT genotypes. The repeated measurement of morphological changes among MsSTTM156 plants and WT over the development was analyzed using the one-way ANOVA. The two-way ANOVA was also used to analyze the expression of IAA transporter and PG genes in those genotypes under Al compared to control conditions. Both one-way and two-way ANOVA were followed by a post hoc Tukey multiple comparison test when significant differences were detected. It means sharing the same letters is not significantly different, with a significance threshold set at *p* ≤ 0.05.

## 5. Conclusions

miR156 is a key regulator of plant growth and response to abiotic stress, including metal toxicity, as shown in Arabidopsis and rice [46,47]. In alfalfa, this miRNA has emerged as a central factor in modulating Al stress response, influencing phenotypic traits, water dynamics, photosynthesis, and molecular pathways.

MsmiR156OE disrupts root development, reduces plant height and biomass, and limits water uptake, as evidenced by decreased relative water content (RWC). This disruption extends to chlorophyll synthesis, impairing photosynthesis and overall plant growth. At the molecular level, MsmiR156OE upregulates auxin transport genes, *MsAUX1* and *MsPIN2*, causing imbalances in IAA distribution that inhibit root elongation, which is in agreement with previous findings [59]. Simultaneously, the downregulation of cell wall-related genes, *MsPG1* and *MsPG4*, increases pectin content and reduces cell wall flexibility, exacerbating Al accumulation and intensifying root damage, as supported by earlier studies [74,75]. These findings are presented in a proposed model (Figure 10) that illustrates miR156 as a master negative regulator of Al response in alfalfa. The model highlights miR156’s central role in regulating phenotypic, physiological, and molecular responses either directly or indirectly through unknown *SPL* genes and their downstream targets. Under Al stress, miR156 modulates the expression of critical genes, including *MsAUX1*, *MsPIN2*, *MsPG1*, and *MsPG4*, leading to root inhibitions and heightened Al susceptibility. Taken together, our results suggest that miR156 is a key negative regulator of Al tolerance in alfalfa.

## Figures and Tables

**Figure 1 plants-14-00958-f001:**
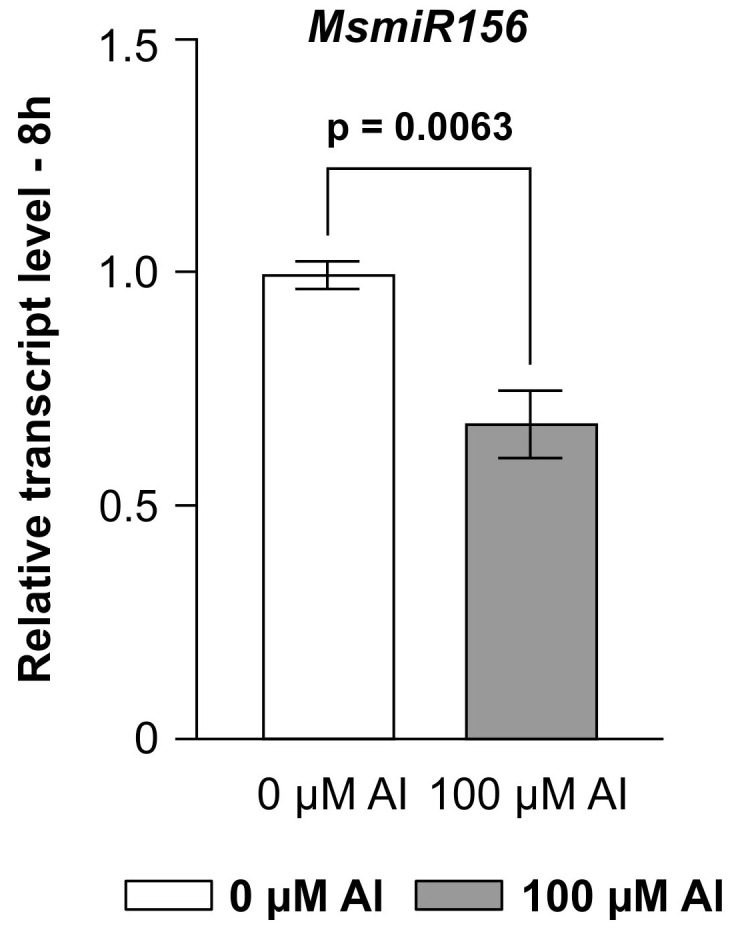
Effect of Al stress on the expression of *MsmiR156* in WT alfalfa roots. Each bar represents the mean of the relative *MsmiR156* transcript level. Error bars represent the standard error of the mean (SEM). A Student’s *t*-test in R (version R-4.2.3) was conducted with n = 4 individual plants. *p* ≤ 0.05 indicates a significant difference between the two groups.

**Figure 2 plants-14-00958-f002:**
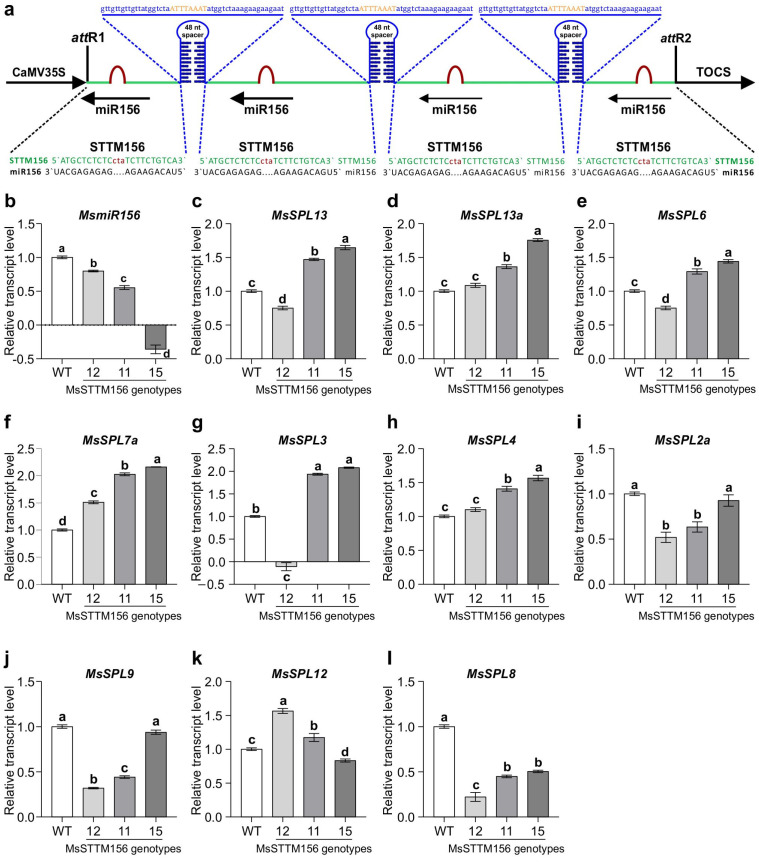
MsSTTM156 (miR156 silencing/knockdown) construct and expression profiles of *MsmiR156* and target *MsSPL* genes in leaves of one-month-old alfalfa plants. (**a**) The MsSTTM156 construct targets various cleavage sites within the MsmiR156 binding regions of *MsSPL* transcripts in alfalfa. The mature sequence of MsmiR156 is shown in black. The sequence of the binding site is shown in green. A mismatched trinucleotide (bulge) is marked in red. The two binding sites separated by a 48-nt spacer create a secondary structure indicated in blue. *Swa*I restriction site is shown in orange in the middle of the spacer. Expression is driven by the CaMV35S promoter and OCS as the transcription terminator. (**b**) Transcript levels of *MsmiR156*, (**c**–**l**) Transcript levels of Ms*SPL13*, *MsSPL13a*, *MsSPL6*, *MsSPL7a*, *MsSPL3*, *MsSPL4, MsSPL2a*, *MsSPL9*, *MsSPL12*, *MsSPL8*. Each bar represents the mean. Error bars indicate the standard error of the mean (SEM). Statistical analysis was performed using one-way ANOVA with n = 4 individual plants for the 11 genes. Significant differences detected by one-way ANOVA in R (version R-4.2.3) were further evaluated using a *post hoc* Tukey multiple comparison test, where means sharing the same letters indicate no significant difference at *p* ≤ 0.05.

**Figure 3 plants-14-00958-f003:**
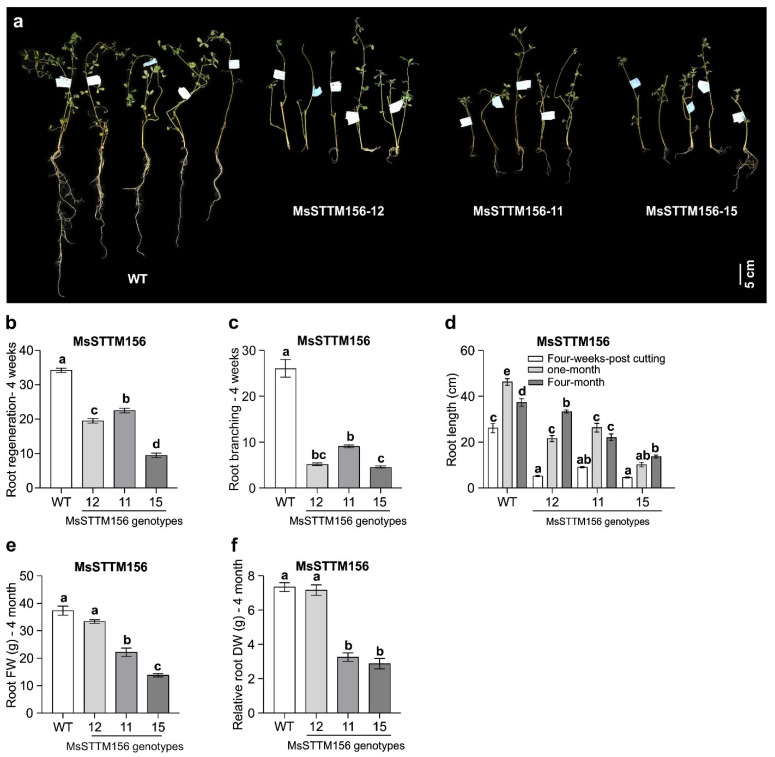
Phenotypic characterization of MsSTTM156 alfalfa plants(miRNA156 silencing/knock down). (**a**) Visual comparison of one-month-old stem cuttings between WT and MsSTTM156 plants (miRNA156 silencing/knock down: MsSTTM156-12, MsSTTM156-11, and MsSTTM156-15), (**b**) root regeneration capacity of one-month-old stem cuttings, (**c**) root branching patterns in one-month-old stem cuttings, (**d**) root length at different developmental stages: one-month-old stem cuttings, one-month post-propagation, and four-month post-propagation, (**e**) fresh weight of roots from four-month-old plants, (**f**) dry weigh of roots from four-month-old plants. Each bar represents the mean. Error bars represent SEM. One-way ANOVA was conducted with n = 4 individual plants. Significant differences detected from the one-way ANOVA in R (version R-4.2.3) were followed by a post hoc Tukey multiple comparison test. Means with the same letters are not significantly difference at a *p* ≤ 0.05.

**Figure 4 plants-14-00958-f004:**
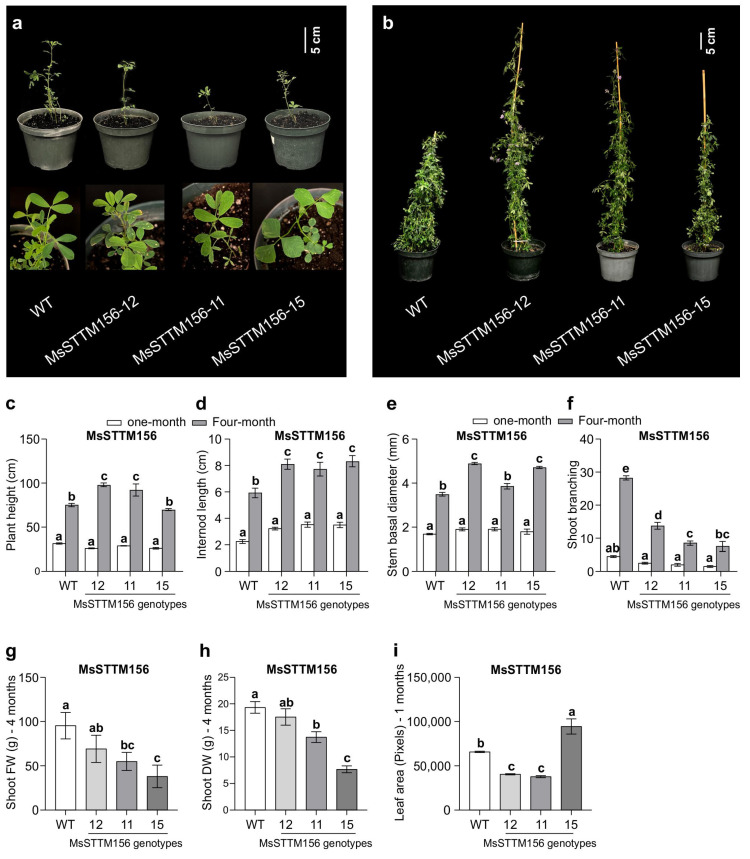
Phenotypic characterization of MsSTTM156 alfalfa plants (miRNA156 silencing/knock down). (**a**) Phenotypic appearance of one-month-old plants, (**b**) phenotypic appearance of four-month-old plants including flowering of MsSTTM156-11 and MsSTTM156-12 at one- and 4-month-old; (**c**) plant height; (**d**) internode length; (**e**) basal stem diameter, and (**f**) shoot branching; as well as (**g**) shoot fresh weight at four months; (**h**) shoot dry weight at four months, and (**i**) leaf area at one-month. Each Error bar represents SEM. One-way ANOVA was conducted with n = 4 individual plants. Significant differences detected from the one-way ANOVA in R (version R-4.2.3) were followed by a post hoc Tukey multiple comparison test. Means with the same letters are not significantly different at a *p* ≤ 0.05.

**Figure 5 plants-14-00958-f005:**
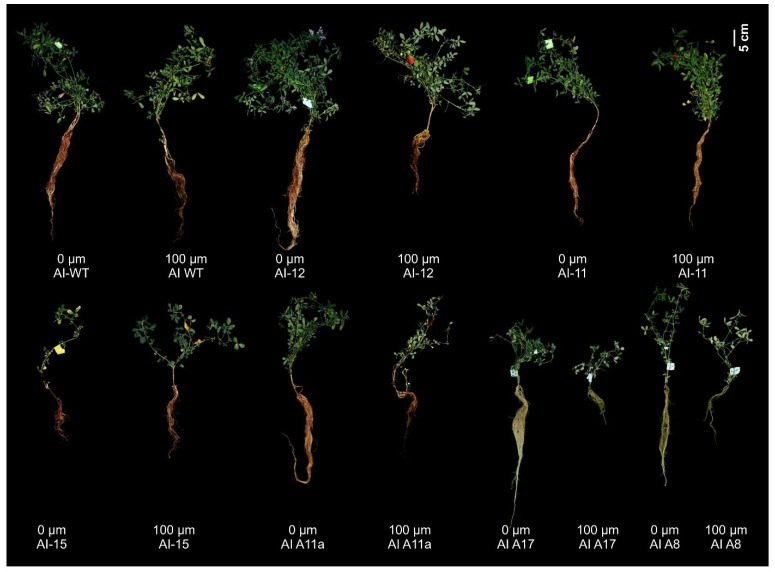
Effects of *MsmiR156* silencing and overexpression on the phenotypic appearance of 63-day-old alfalfa plants under Al stress. MsSTTM156 genotypes (miRNA156 silencing/knock down: 12, 11, and 15), MsmiR156OE genotypes (A11a, A17, and A8), and WT plants. Each genotype was either left untreated (control, 0 µM Al) or exposed to Al treatment (100 µM Al). The control group is denoted as 0 µM Al-WT, 0 µM Al-12, 0 µM Al-11, 0 µM Al-15, 0 µM Al-A11a, 0 µM Al-A17, and 0 µM Al-18. The Al-treated group, subjected to 100 µM Al, includes 100 µM Al-WT, 100 µM Al-12, 100 µM Al-11, 100 µM Al-15, 100 µM Al-A11a, 100 µM Al-A17, and 100 µM Al-18. Phenotypic changes, recorded 14 days after Al treatment, included root shortening and browning caused by increased acidity under Al stress. Scale bar = 5 cm.

**Figure 6 plants-14-00958-f006:**
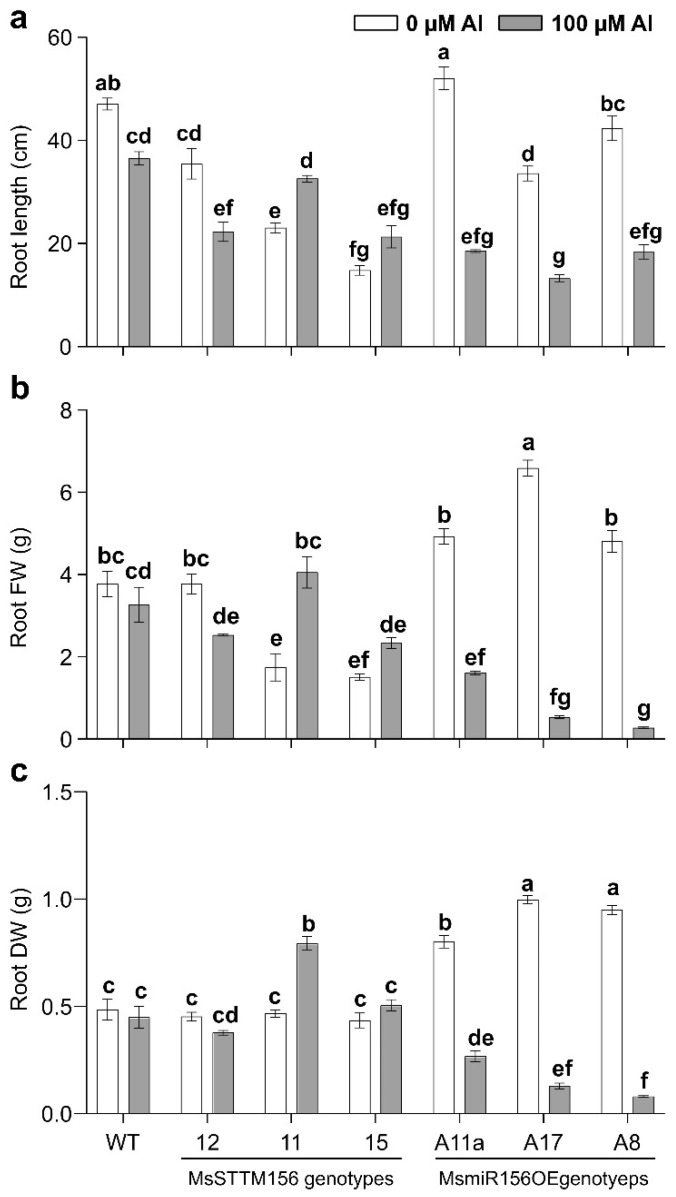
Effects of *MsmiR156* silencing and overexpression on alfalfa’s phenotypic response to 14 days of Al stress. (**a**) root length of MsSTTM156, MsmiR156OE, and WT exposed; (**b**) root fresh weight, (**c**) root dry weight; Each bar indicates a biological mean. Error bars represent SEM. A two-way ANOVA was conducted across all treatments and genotypes, with n = 4 individual plants. Significant results detected from the two-way ANOVA in R (version R-4.2.3) were followed by a post hoc Tukey multiple comparison test. Means with the same letters are not significantly different at a *p* ≤ 0.05.

**Figure 7 plants-14-00958-f007:**
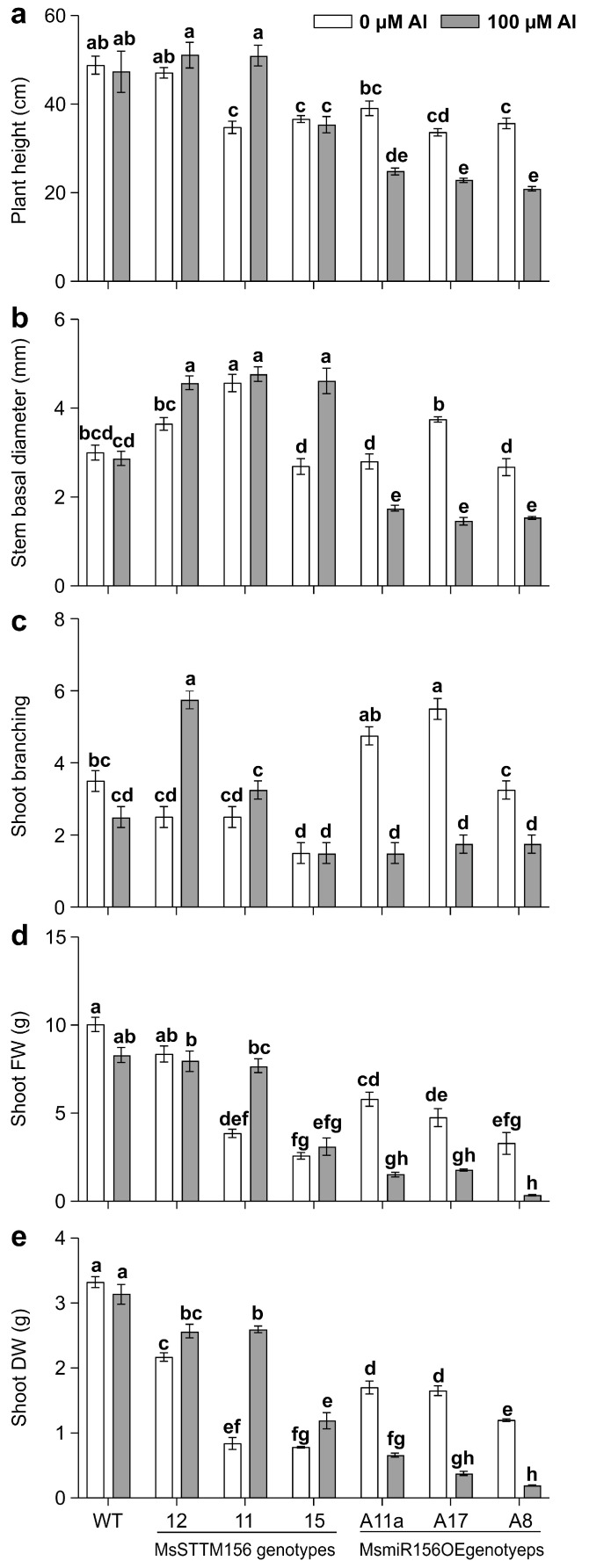
Effects of Ms*miR156* silencing and overexpression on alfalfa’s phenotypic response to Al stress. (**a**) plant height of MsSTTM156 (11, 12, 15), MsmiR156OE (A11a, A17, A8), and WT plants exposed to Al stress for 14 days, (**b**) basal stem diameter, (**c**) shoot branching, (**d**) shoot fresh weight, (**e**) shoot dry weight. Each bar indicates the mean. Error bars represent SEM. A two-way ANOVA was conducted across all treatments and genotypes, with n = 4 individual plants. Significant differences detected from the two-way ANOVA in R (version R-4.2.3) were followed by a post hoc Tukey multiple comparison test. Means with the same letters are not significantly different at *p* ≤ 0.05.

**Figure 8 plants-14-00958-f008:**
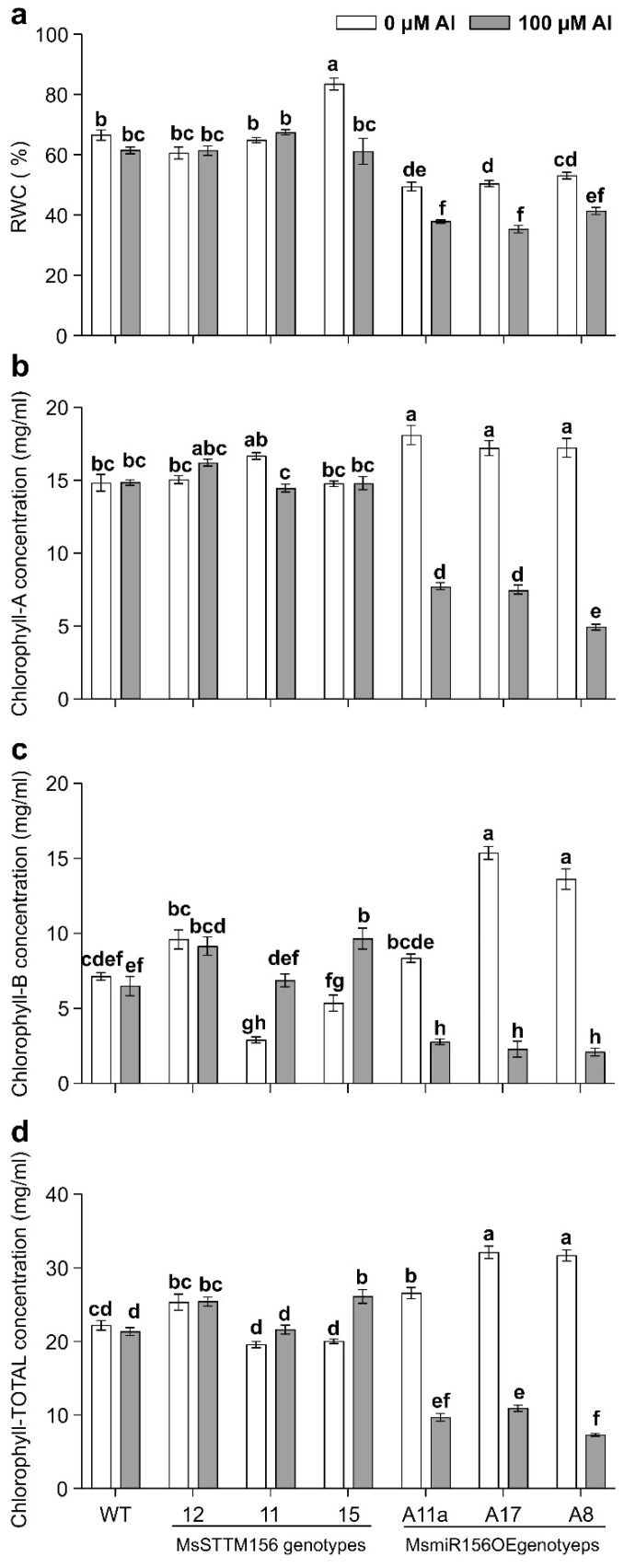
Effects of *MsmiR156* silencing (MsSTTM156) and overexpression (MsmiR156OE) on alfalfa’s physiological response to Al stress. (**a**) leaf water content, (**b**) chlorophyll-A, (**c**) chlorophyll-B, (**d**) chlorophyll-TOTAL; Each bar plot indicates the mean. Error bars represent SEM. A two-way ANOVA was conducted across all treatments and genotypes, with n = 4 individual plants. Significant differences detected from the two-way ANOVA in R (version R-4.2.3) were followed by a post hoc Tukey multiple comparison test. Means with the same letters are not significantly different at a *p*-value ≤ 0.05.

**Figure 9 plants-14-00958-f009:**
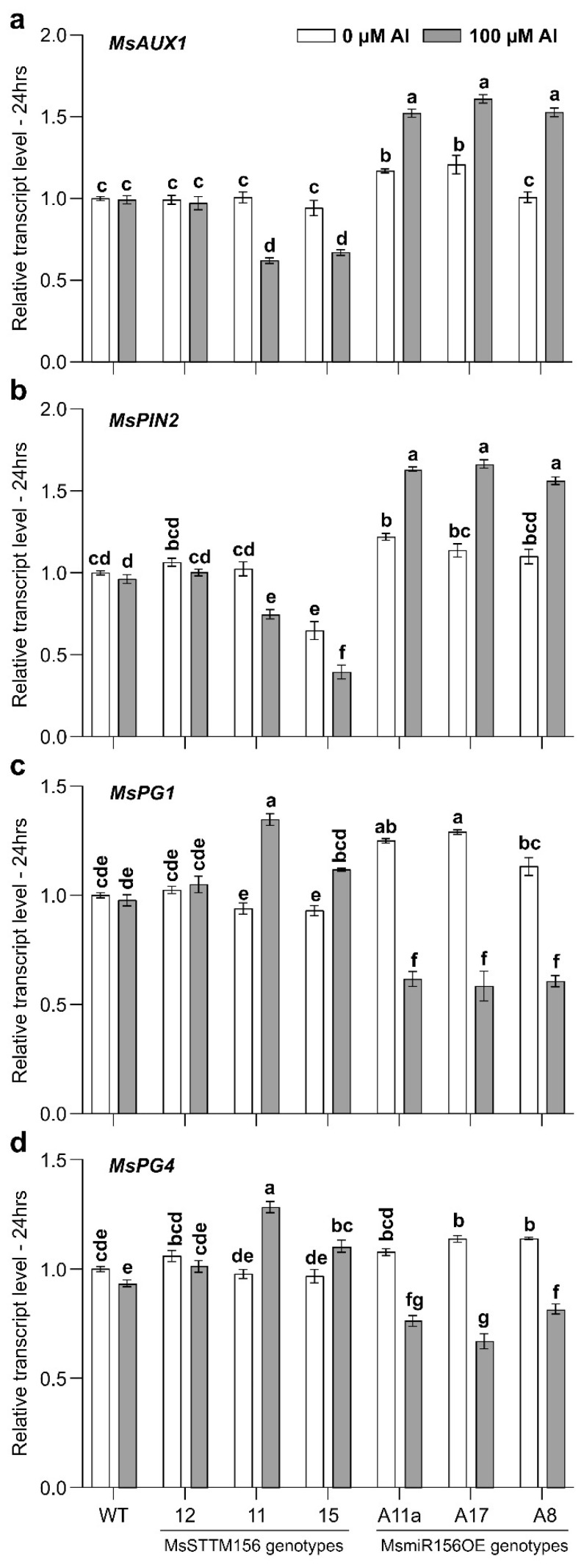
Expression patterns of *MsAUX*1, *MsPIN2*, *MsPG*1, and *MsPG4* in the root tips of MsSTTM156, MsmiR156OE, and WT plants exposed to 100 µM Al for 24 h. (**a**) *MsAUX1*, (**b**) *MsPIN2*, (**c**) *MsPG1*, (**d**) *MsPG4*. Each bar plot indicates the mean. Error bars represent SEM. A two-way ANOVA was conducted across all treatments and genotypes, with n = 4 individual plants. Significant differences detected from the two-way ANOVA in R (version R-4.2.3) were followed by a post hoc Tukey multiple comparison test. Means with the same letters are not significantly different at a *p*-value ≤ 0.05.

**Figure 10 plants-14-00958-f010:**
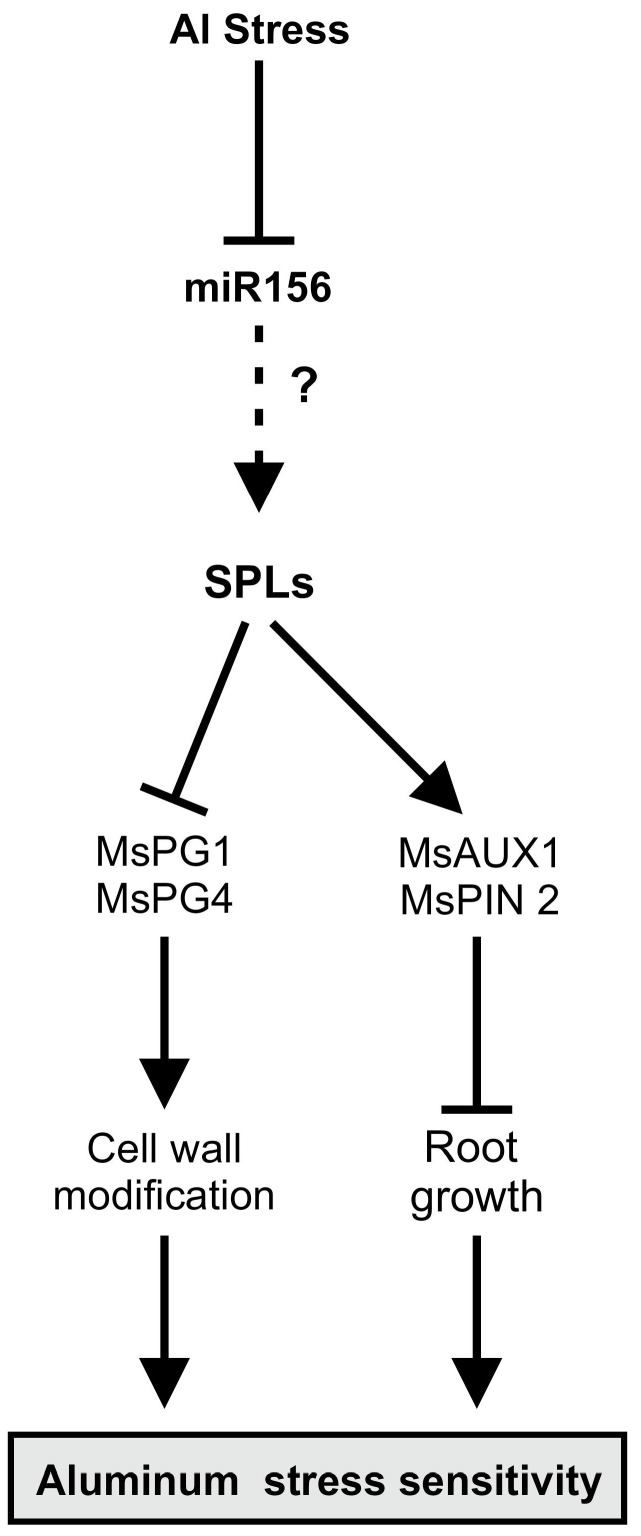
Schematic representation of miRNA156-based alfalfa Al tolerance model negative regulation. Solid lines represent an experimentally confirmed mechanism, while broken lines are hypothesized functions. Arrow heads indicate positive regulation, while line heads indicate negative regulation. Question mark (?) indicates hypothetical uncharacterized steps. Overall, Al stress causes a reduction in *miRNA156* expression, leading to a release of *SPL* gene repression. SPLs function to inhibit MsPG1 and MsPG4, leading to enhanced cell wall modifications. Simultaneously, SPLs function to enhance MsAUX1 and MsPIN2 expression, leading to altered auxin patterns and disrupting root growth. Both cell wall modifications and reduced root growth contribute to overall Al stress sensitivity.

## Data Availability

The datasets generated during and/or analyzed during the current study are available from the corresponding author upon reasonable request.

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
