# Peer review of "miR156 Is a Negative Regulator of Aluminum Response in Medicago sativa"

_plants, 2025, doi:10.3390/plants14060958_

Round 1

Reviewer 1 Report

Comments and Suggestions for Authors

plants-3486429 “miR156 is a negative regulator of aluminum response in Medicago sativa”.

Studying miRNA functions in plants is a highly relevant topic, since miRNAs play important roles in plants, but there is still scarce information on miRNA functions. This study plants-3486429 provides new information about the functions of miRNA in the plant's response to aluminum. The research is extensive, with a lot of experiments, and has been carried out thoroughly. The results are described in detail, statistical analysis was performed and described at a high-level. There are several comments that need to be addressed.

- Introduction. Lines 64-66. It is necessary to clarify and provide information what aspects and processes SPL transcription factors regulate? Please add about 2-3 sentences on this issue here.

- Introduction, lines 99-109. It would be good to mention in the Introduction what are other than STTM popular methods of miRNA downregulation? Why STTM has been chosen and not other method.

- Results. Please clarify why this particular aluminium concentration has been chosen? 100 µM. Why are you using only one Al concentration in your investigation? Did you test several ones? It is necessary to include a note on this in the manuscript (e.g in section 2.1 and 2.4.).

- Using only one Al concentration is particularly questionable, because 100 µM of Al did not cause a significant effect on wild-type plant growth and physiological parameters, including Root FW, Root DW (Fig. 6), plant height, stem basal diameter, shoot branching, shoot FW/DW, chlorophyll concentration. At the same time the authors report in section 2.6 that “Al stress is a well documented abiotic challenge that disrupts water uptake [50,51]  and significantly reduces chlorophyll concentration [19].”Affecting growth of control plants is an important issue when setting up experiments of such type.

- Figure 3 legend. Correct “four-week-old stem cuttings” to one-month-old stem cuttings” (as in other figures).

- Figure legends. It would be helpful for readers to clarify MsSTTM156 plants (miRNA156 silencing/knock down) in the names of Fig. 2, Fig. 3, etc in their legends.  

- Abstract, line 19. Correct “..we utilized short tandem target mimic (STTM) to silence…” to “we utilized short tandem target mimic (STTM) method to silence…”.

- Abstract,  it is necessary to provide full names for  IAA, MsPG1, MsPG4, MsAUX1  and  MsPIN2 in Abstract.

Author Response

REVIEWER 1
Overall outlook: Studying miRNA functions in plants is a highly relevant topic, since miRNAs play important roles in plants, but there is still scarce information on miRNA functions. This study plants-3486429 provides new information about the functions of miRNA in the plant's response to aluminum. The research is extensive, with a lot of experiments, and has been carried out thoroughly. The results are described in detail, statistical analysis was performed and described at a high-level. There are several comments that need to be addressed.

Comment 1: Introduction. Lines 64-66. It is necessary to clarify and provide information what aspects and processes SPL transcription factors regulate? Please add about 2-3 sentences on this issue here.
Response 1: The lines 64-66 were revised to include three sentences to clarify and provide information abouts aspects and processes that SPL transcription factors regulate in plants (see lines 67-73).

Comment 2: Introduction, lines 99-109. It would be good to mention in the Introduction what are other than STTM popular methods of miRNA downregulation? Why STTM has been chosen and not other method.
Response 2: As per the reviewer’s suggestion, the lines 99-109 were revised to include other methods used to downregulate miRNA and why STTM was chosen to conduct our study (lines 118-125).
Comment 3: Results. Please clarify why this particular aluminium concentration has been chosen? 100 µM. Why are you using only one Al concentration in your investigation? Did you test several ones? It is necessary to include a note on this in the manuscript (e.g in section 2.1 and 2.4.). 
Response 3: The selection of 100 µM Al was based on prior studies evaluating Al stress in Medicago sativa, where a range of concentrations (0, 50, 100, 200, and 300 µM) were tested (Wang et, 2016, ref # 59 in manuscript).  Wang et al. (2016) demonstrated that root elongation inhibition was positively correlated with Al concentration, with a significant reduction observed at 100 µM after 48 hours. While 50 µM Al induced minimal stress symptoms, 200 and 300 µM were highly toxic, leading to severe root growth inhibition. Therefore, 100 µM was chosen as the optimal concentration to study miR156-mediated responses while avoiding excessive stress damage. To clarify this, we have revised Section 2.1 as follows: “The severity of Al toxicity in plants is influenced by multiple factors, including Al concentration, exposure duration, developmental stage, growth conditions, and plant species[21].“Previous research showed that 100 µM Al was optimal for evaluating Al-induced stress, and its impact on plant growth in Medicago sativa [59], and thus this concentration was used in our study”. This text can be found in lines 144-147. Since this information is added to section 2.1 (Results), we  do not repeat it in section 2.4 (Methods). We  further evaluated three concentrations, 0, 100,  200 µM in our study on WT plants and found that 100 µM to be  optimal, while 200 caused sever txocicty to the plants (see Fig.4S on page # 6 of the supplementary file).
Comment 4: Using only one Al concentration is particularly questionable, because 100 µM of Al did not cause a significant effect on wild-type plant growth and physiological parameters, including Root FW, Root DW (Fig. 6), plant height, stem basal diameter, shoot branching, shoot FW/DW, chlorophyll concentration. At the same time the authors report in section 2.6 that “Al stress is a well documented abiotic challenge that disrupts water uptake [50,51]  and significantly reduces chlorophyll concentration [19].”Affecting growth of control plants is an important issue when setting up experiments of such type.
Response 4: As clarified in response to the previous comment, the selection of 100 µM Al was based on prior studies that  identified it as an optimal concentration for studying Al toxicity in Medicago sativa. This study specifically investigates the role of miR156 in  regulating  Al stress responses. The absence of significant growth inhibition in WT plants suggests a potential dose-dependent effect of miR156, where its silencing or overexpression alters plant sensitivity to Al stress. While WT plants exhibited resilience at this concentration in these parameters, transgenic plants displayed distinct phenotypic and physiological responses, supporting the hypothesis that miR156 plays a crucial regulatory role in Al stress response.
Comment 5: Figure 3 legend. Correct “four-week-old stem cuttings” to one-month-old stem cuttings” (as in other figures).
Response 5: As per the reviewer’s comment, the term “four-week-old stem cuttings”  has been corrected to “ one-month-old stem cuttings”  in  the legend of Figure 3,  and all relevant figures to ensure consistency throughout the manuscript. This revision has been applied to  lines 251, 253, and 254. 

Comment 6: Figure legends. It would be helpful for readers to clarify MsSTTM156 plants (miRNA156 silencing/knock down) in the names of Fig. 2, Fig. 3, etc in their legends.  

Response 6: The figure legends were revised to clarify MsSTTM156 plants names included  miR156 silencing/knockdown  in Fig.2, Fig.3, Fig4, and Fig5. These revision have been applied to lines 183, 250, 252, 263, and 321

Comment 7: Abstract, line 19. Correct “..we utilized short tandem target mimic (STTM) to silence…” to “we utilized short tandem target mimic (STTM) method to silence…”.
Response 7: The sentence has been revised as suggested: "We utilized the short tandem target mimic (STTM) method to silence miR156 in alfalfa. Specifically, line 19 was altered to include this change. 

Comment 8: Abstract, it is necessary to provide full names for IAA, MsPG1, MsPG4, MsAUX1 and MsPIN2 in Abstract.
Response 8: As per the reviewer’s suggestion, we have included the full names for IAA, MsPG1, MsPG4, MsAUX1 and MsPIN2 in the abstract. The revised text included the following indole-3-acetic acid (IAA), polygalacturonase 1 (MsPG1), polygalacturonase 4 (MsPG4), Auxin transporter-like protein 1(MsAUX1), and Auxin efflux carrier components 2 (MsPIN2). The revised names have been applied to lines 29, 30, 31, and 32.

Reviewer 2 Report

Comments and Suggestions for Authors

This work focuses on the role of miRNAs on Medicago responses to aluminum stress. Transgenic plants with repressed (STTM technology) or overexpressed mi-RNA156 are used as a tool for evaluation of the individual effects on its selected SPL targets. Although the work clearly presents interesting and valuable results, it lacks some serious aspects.

  1. The authors work with clonally propagated experimental plants, but do not present any analyses of transgenic plants under- or overexpressing miRNA156.
  2. The effects of suppressing miRNA156 levels on the morpho-physiology of transgenic plants are presented, but these are not presented in the same way for miRNA156-overexpressing plants. The Goldilocks effect supported by data suggests that optimal levels of this molecule are crucial - both too little and too much lead to adverse effects. Therefore, without overexpression data, we can only see a partial picture.
  3. A nice data set is presented on the impact of both under- and overexpression of miRNA156 on SPLs when grown in the presence of aluminium. It is correct that the authors used 2-ANOVA (I would also recommend non-linear regression or quadratic term modeling given my above comment 2). However, the results are presented without comparative interpretations based on the contrast between under- and overexpression of miRNA156.
  4. On what basis did the authors select miRNA156 targets for study?
  5. We already have knowledge about the role of this regulatory molecule for other metals - (Cd, Zn..); it would make sense to introduce these more in the introduction
  6. Fig S3 presents per se only data for the number of flowers; the legend should therefore not combine these data with information on the period of blooming, which is not presented anywhere
  7. Fig. 5 also indicates changes in the synthesis of substances causing root coloration; literature relevant to the experiment is available on this phenomenon

Author Response

REVIEWER 2

Overall outlook: This work focuses on the role of miRNAs on Medicago responses to aluminum stress. Transgenic plants with repressed (STTM technology) or overexpressed mi-RNA156 are used as a tool for evaluation of the individual effects on its selected SPL targets. Although the work clearly presents interesting and valuable results, it lacks some serious aspects. 

Comment 1: The authors work with clonally propagated experimental plants, but do not present any analyses of transgenic plants under- or overexpressing miRNA156

Response 1: This study specifically focused on the detailed analysis of miR156 silencing/knockdown (STTM) transgenic plants, as they were generated as part of this research. The miR156 overexpression (miR156OE) plants we used in this study were previously generated and characterized by Aung et al., 2015 in our lab, as cited in the Methods section (Section 4.1, lines 744-746).  We respectfully submit that it would not be appropriate for us to repeat work on miR156 overexpression that has already been published.

Comment 2: The effects of suppressing miRNA156 levels on the morpho-physiology of transgenic plants are presented, but these are not presented in the same way for miRNA156-overexpressing plants. The Goldilocks effect supported by data suggests that optimal levels of this molecule are crucial - both too little and too much lead to adverse effects. Therefore, without overexpression data, we can only see a partial picture.

Response 2: We are not sure what the reviewer is referring to in this case, because our morpho-physiological characterization clearly included use of WT, miR156 overexpression (miR156OE) and miR156 silencing (STTM) plants. So, we used both overexpression and silencing plants in addition to the WT control.  Please also see our response to the previous comment 2.

Comment 3: A nice data set is presented on the impact of both under- and overexpression of miRNA156 on SPLs when grown in the presence of aluminium. It is correct that the authors used 2-ANOVA (I would also recommend non-linear regression or quadratic term modeling given my above comment 2). However, the results are presented without comparative interpretations based on the contrast between under- and overexpression of miRNA156.

Response 3: We thank the reviewer for their suggestion. A two-way ANOVA was chosen as the most suitable statistical method for this study because it allows for the simultaneous evaluation of two independent categorical factors. For example genotype (WT vs. MsSTTM156 or WT vs. MsmiR156OE) and treatment (control vs. aluminum stress) and  their interaction on physiological and morphological outcomes. This method ensures that each factor's effect is independently assessed while also identifying potential interactions between genotypes and treatments. In contrast, non-linear regression and quadratic modeling are typically used to describe continuous relationships, such as dose-response curves, where the predictor variable varies along a gradient. However, in this study, Al treatment was applied as a categorical variable (presence or absence of Al stress), rather than as a continuous range of concentrations, making regression-based methods less appropriate. Additionally, a two-way ANOVA provides comparisons between experimental groups, allowing for clear statistical distinctions between WT and transgenic plants under different conditions.

Comment 4: On what basis did the authors select miRNA156 targets for study?

Response 4: We have added information in section 2.2. (lines 166-168) to indicate the reasoning behind analyzing the expression of the SPL genes in MsSTTM156 plants. While miR156 targets SPL genes, the primary focus of this study was to investigate the broader role of miR156 in Al stress responses rather than detailed SPL regulation.

Comment 5: We already have knowledge about the role of this regulatory molecule for other metals - (Cd, Zn.); it would make sense to introduce these more in the introduction

Response 5: As per the reviewer's recommendation, we have expanded the introduction to include details on the role of miR156 in regulating Zn, emphasizing its function as a key regulator of mineral nutrient homeostasis (lines 86-89). Additionally, we have incorporated the role of miR156 in Cd regulation within the broader context of its involvement in abiotic stress responses (lines 104-115). 
Comment 6: Fig S3 presents per se only data for the number of flowers; the legend should therefore not combine these data with information on the period of blooming, which is not presented anywhere.

Response 6: We thank the reviewer for this observation.  Accordingly, Fig S3 legend has been revised to accurately reflect the data presented, focusing only on flower number and eliminating the information on the period to blooming time or flowering (Page 5 of Suplmentary file). 

Comment 7: Fig. 5 also indicates changes in the synthesis of substances causing root coloration; literature relevant to the experiment is available on this phenomenon

Response 7: As per the reviewer’s suggestion, we have incorporated literature-supported details explaining the root coloration changes observed under Al stress into Figure 5 (lines 326-327), the results section (section 2.4, lines 290–292), and the discussion (section 3.6, lines 583–599). These additions specifically address the synthesis and accumulation of phenolic compounds and oxidative stress mechanisms, clarifying the biochemical basis of root browning in our experimental plants.

Reviewer 3 Report

Comments and Suggestions for Authors

In this study, authors investigated the regulatory role of miR156 in Al stress response in Medicago sativa. They found through overexpression and gene silencing techniques that miR156 negatively regulates the resistance of alfalfa under Al stress. Under Al stress, the growth of plants overexpressing miR156 was inhibited, and their biomass decreased. Moderate silencing of miR156, however, could increase the dry weight of roots and enlarge the basal diameter of stems. Molecular-level studies revealed that silencing miR156 can regulate the transcription of cell wall-related genes associated with aluminum tolerance and auxin transport-related genes. These results reveal the multi-layered role of miR156 in regulating the aluminum tolerance of alfalfa, providing important insights into the genetic strategies of alfalfa in response to aluminum stress. The methods and results are acceptable. There are some essential problems should be addressed by authors, which are listed below.

  1. Please check for writing errors throughout the text. For example, in figure 10 “miR56” should change to “miR156”.
  2. L23, “significant changes?”. Authors need to clearly state in the abstract whether the effect of miR156 on plant growth under normal conditions is positive or negative.
  3. L97-L107, I think the content here can be deleted.
  4. In 2.1, Why did the authors choose a concentration of 100 μM for the determination? Are there any references for this concentration selection? Moreover, is the expression level of miR156 the same at lower or higher concentrations? Does the stress duration affect the expression of miR156? It is recommended to add RT-qPCR experiments for further clarification.
  5. In Figure 2g, the expression level of MsSPL3 in the STTM156-12 line is much lower than that in others. Why? The authors did not explain this in the results.
  6. It is recommended to use the psRNATarget (https://www.zhaolab.org/psRNATarget/) to predict whether miR156 interacts with SPLs.
  7. The content of the Discussion section is too lengthy. Please trim it down as appropriate. For example, Sections 3.2 - 3.4, which focus on the effects of miR156 on plant growth under normal conditions, are recommended to be integrated into one paragraph. L591-610: Here, the authors have made extensive descriptions only to conclude that the reduction in shoot biomass in miR156-OE plants is due to root-related effects. I think this is of little significance. The authors should directly approach the issue from a molecular level and conclude that miR156OE upregulates the expression of MSAUX1 and MSPIN2, leading to an imbalance in IAA distribution, inhibiting root elongation, and thus affecting shoot biomass, and so on. There are many such instances in the manuscripts.
  8. It is recommended that the authors strengthen the discussion on the interactions among miR156-SPLs-critical genes. For example, explore whether there is a direct interaction (cleavage or translation) between miR156 and SPLs, and how SPLs regulate the expression of critical genes. The authors should make reasonable speculations about the underlying mechanisms.
Comments on the Quality of English Language

See above

Author Response

REVIEWER 3
Overall outlook: In this study, authors investigated the regulatory role of miR156 in Al stress response in Medicago sativa. They found through overexpression and gene silencing techniques that miR156 negatively regulates the resistance of alfalfa under Al stress. Under Al stress, the growth of plants overexpressing miR156 was inhibited, and their biomass decreased. Moderate silencing of miR156, however, could increase the dry weight of roots and enlarge the basal diameter of stems. Molecular-level studies revealed that silencing miR156 can regulate the transcription of cell wall-related genes associated with aluminum tolerance and auxin transport-related genes. These results reveal the multi-layered role of miR156 in regulating the aluminum tolerance of alfalfa, providing important insights into the genetic strategies of alfalfa in response to aluminum stress. The methods and results are acceptable. There are some essential problems should be addressed by authors, which are listed below.

Comment 1: Please check for writing errors throughout the text. For example, in figure 10 “miR56” should change to “miR156”.

Response 1: The entire manuscript has been carefully reviewed and all typographical errors corrected, including the revision of "miR56" to "miR156" in Figure 10.

Comment 2: L23, “significant changes?”. Authors need to clearly state in the abstract whether the effect of miR156 on plant growth under normal conditions is positive or negative.
Response 2: Line 23 has been revised and clearly state that silencing miR156 induces significant negative changes in plant growth under normal conditions.  Text on line 23 was modified to include this change in the abstract. 
Comment 3: L97-L107, I think the content here can be deleted.
Response 3: While we would be willing to remove text in lines L97-107, we prefer to keep it for now, as in our view it provides essential background information about studying the loss of function of miRNAs using STTM and other approaches, and why STTM was chosen over the others, which was suggested for addition by reviewer # 1. However, if the reviewer and editor prefer deleting it, we will be happy to do so.

Comment 4: In 2.1, Why did the authors choose a concentration of 100 μM for the determination? Are there any references for this concentration selection? Moreover, is the expression level of miR156 the same at lower or higher concentrations? Does the stress duration affect the expression of miR156? It is recommended to add RT-qPCR experiments for further clarification.

Response 4: The question of Al concentration was also raised by Reviewer 2, and we have addressed it by the fact that this concentration was based on the result of a previous study by Wang et al, 2016 (reference #59 in the manuscript), which found 100 µM to be optimal to study Al stress.  See text in lines lines 144-147  (section 2.1). The expression of miR156 was down reuregualted with concentrations as low as 10 µM in some species, and with 100 µM in alfalfa. It depens on the plant species. As described in the Methods section, we used RT-qPCR to assess expression of miR156 and SPL genes.While we agree with the reviewer that studies on the effect of duration of Al stress on the expression levels are important, we respectfully submit that they are beyond the scope of the current study.  

Comment 5: In Figure 2g, the expression level of MsSPL3 in the STTM156-12 line is much lower than that in others. Why? The authors did not explain this in the results.

 Response 5: We appreciate the reviewer’s insightful comment. To clarify the notably lower expression of MsSPL3 in the STTM156-12 line shown in Figure 2g, we have revised the figure and provided a clear explanation in the Results section 2.2 (lines 175-176) and discussion section 3.2 ( lines 501-505).

Comment 6: It is recommended to use the psRNATarget(https://www.zhaolab.org/psRNATarget/) to predict whether miR156 interacts with SPLs.

Response 6: We appreciate the reviewer’s recommendation, but the focus of the study is only on the role of miR156 (and not its target SPL genes)  in response to Al stress. Future studies will definitely include analysis miR156-SPL network in regulating Al response, in which  performing psRNATarget predictions for miR156 binding sites on SPL genes would be highly valuable.  It should be noted that for many of the target SPL genes in alfalfa, we have already determined the miR156 target sites in some of our previous studies.These studies include: 

1.    Aung B, Gruber M, Amyot L, Omari K, Bertrand A and Hannoufa A (2015) MicroRNA156 as a promising tool for alfalfa improvement.  Plant Biotech J. (2015) 13, pp. 779–790
2.    Gao R., Austin R.S., Amyot L., Hannoufa A. (2016) Comparative transcriptome investigation of global gene expression changes caused by miR156 overexpression in Medicago sativa. BMC Genomics 17:658
3.    Feyissa BA, Amyot L, Nasrollahi V, Papadopoulos Y,  Kohalmi S, Hannoufa A (2021) Involvement of the miR156/SPL module in flooding response in Medicago sativa. Scientific Reports. http://doi.org/10.1038/s41598-021-82450-7

Comment 7: The content of the Discussion section is too lengthy. Please trim it down as appropriate. For example, Sections 3.2 - 3.4, which focus on the effects of miR156 on plant growth under normal conditions, are recommended to be integrated into one paragraph. L591-610: Here, the authors have made extensive descriptions only to conclude that the reduction in shoot biomass in miR156-OE plants is due to root-related effects. I think this is of little significance. The authors should directly approach the issue from a molecular level and conclude that miR156OE upregulates the expression of MSAUX1 and MSPIN2, leading to an imbalance in IAA distribution, inhibiting root elongation, and thus affecting shoot biomass, and so on. There are many such instances in the manuscripts
Response 7: As per the reviewer’s suggestion, we have condensed sections 3.2-3.4 to briefly integrate the effects of miR156 on plant growth under normal conditions. Section 3.2 was shortened from 14 to 12 lines (lines 515–527), Section 3.3 from 16 to 12 lines (lines 529–541), and Section 3.4 from 20 to 13 lines (lines 543–557). Additionally, to directly address the reviwer’s recommendation, lines 591-610 have been revised (see lines 644–659) to emphasize the molecular basis of biomass reduction. Specifically, we clarified that miR156OE plants show upregulated expression of auxin transport genes (MsAUX1 and MsPIN2), leading to disrupted IAA distribution, inhibited root elongation, and consequently reduced shoot biomass.

Comment 8: It is recommended that the authors strengthen the discussion on the interactions among miR156-SPLs-critical genes. For example, explore whether there is a direct interaction (cleavage or translation) between miR156 and SPLs, and how SPLs regulate the expression of critical genes. The authors should make reasonable speculations about the underlying mechanisms.

Response: Again, as we indicated in response to the reviewer’s previous comment #6, the focus of this study is solely to investigate the role of miR156 (and not its target SPL genes) in Al response in alfalfa.  Therefore, we think that any speculation about the role (s) of SPLs in Al response would be purely speculative in the absence of concrete data.  Note that in some of our previous studies we already reported on the  target SPL genes that were silenced via transcript cleavage by miR156 under vatrious conditions, as indicated in our response to the reviewer’s aforementioned comment (#6), including the cited references.

Round 2

Reviewer 2 Report

Comments and Suggestions for Authors

I appreciate the authors' efforts in addressing my comments and for their precise edits and explanations. The additional information provided has greatly improved the clarity of the manuscript. I also accept their arguments regarding the statistical methods used in the study.

My initial concerns in points 1 and 2 of my original review  stemmed from unclear/insufficient reference to the experimental plants from the previous study (Aung et al., 2014). To enhance clarity, I recommend that the authors add a brief statement at the beginning of the discussion section on that the miR156-overexpressing plants have already (partly) been characterized in the aforementioned study. Additionally, they should clarify that the data obtained from clones of these plants (either in this or the previous study) are compared with the findings from plants with suppressed miR156 expression (or something similar).

Regarding point 2, I originally highlighted the need for a more detailed confrontation/explanation of the differences in the effects of overexpression and suppression of miR156 on various parameters, referring to the so-called "Goldilocks effect." In section 3.4, the authors admit that the mechanism of the similar effect of silencing and miR156 overexpression on flowering delay is not yet elucidated and requires further research; they did not provide a more detailed discussion of possible hypotheses or mechanisms that could explain such an effect (e.g., via regulatory loops, epigenetic mechanisms, or interactions with other flowering factors). In other sections,  they have provided available insights into possible explanations for differences between the impact of under- and overexpression. It seems they have explored explanations where the data allowed. Nonetheless, the manuscript would benefit from a brief summary explicitly comparing the effects of miR156 overexpression and suppression.

Author Response

REVIEWER 2

Overall outlook: I appreciate the authors' efforts in addressing my comments and for their precise edits and explanations. The additional information provided has greatly improved the clarity of the manuscript. I also accept their arguments regarding the statistical methods used in the study.

Comment 1: My initial concerns in points 1 and 2 of my original review stemmed from unclear/insufficient reference to the experimental plants from the previous study (Aung et al., 2014). To enhance clarity, I recommend that the authors add a brief statement at the beginning of the discussion section on that the miR156-overexpressing plants have already (partly) been characterized in the aforementioned study. Additionally, they should clarify that the data obtained from clones of these plants (either in this or the previous study) are compared with the findings from plants with suppressed miR156 expression (or something similar).

Response 1: As per the reviewer’s suggestion, we have added a statement at the beginning of the discussion section to clarify that the miR156-overexpressing plants used in this study were initially characterized in the study by Aung et al. 2015[reference# 35]. We also clarified that the data obtained from these plants were compared with findings from plants with suppressed miR156 expression. These changes can be found in lines 485-490.

Comment 2: Regarding point 2, I originally highlighted the need for a more detailed confrontation/explanation of the differences in the effects of overexpression and suppression of miR156 on various parameters, referring to the so-called "Goldilocks effect." In section 3.4, the authors admit that the mechanism of the similar effect of silencing and miR156 overexpression on flowering delay is not yet elucidated and requires further research; they did not provide a more detailed discussion of possible hypotheses or mechanisms that could explain such an effect (e.g., via regulatory loops, epigenetic mechanisms, or interactions with other flowering factors). In other sections,  they have provided available insights into possible explanations for differences between the impact of under- and overexpression. It seems they have explored explanations where the data allowed. Nonetheless, the manuscript would benefit from a brief summary explicitly comparing the effects of miR156 overexpression and suppression.

Response 2: As per the reviewer’s suggestion, we have provided a more detailed discussion of possible mechanisms behind the similar flowering delay observed with both miR156 silencing and overexpression plants, including potential regulatory loops and interactions with other flowering factors (lines 565-574). We also included a summary explicitly comparing the effects of miR156 overexpression and suppression on flowering, which can be found in lines 574-575.

Reviewer 3 Report

Comments and Suggestions for Authors

Authors have addressed my concerns.

Author Response

Comment 1: Authors have addressed my concerns.

Response 1: We thank the reviewer for helping us improve our work.